# A Unified View on Graph Neural Networks as Graph Signal Denoising

## Abstract

Graph Neural Networks (GNNs) have risen to prominence in learning representations for graph structured data. A single GNN layer typically consists of a feature transformation and a feature aggregation operation. The former normally uses feed-forward networks to transform features, while the latter aggregates the transformed features over the graph. Numerous recent works have proposed GNN models with different designs in the aggregation operation. In this work, we establish mathematically that the aggregation processes in a group of representative GNN models including GCN, GAT, PPNP, and APPNP can be regarded as (approximately) solving a graph denoising problem with a smoothness assumption. Such a unified view across GNNs not only provides a new perspective to understand a variety of aggregation operations but also enables us to develop a unified graph neural network framework UGNN. To demonstrate its promising potential, we instantiate a novel GNN model, ADA-UGNN, derived from UGNN, to handle graphs with adaptive smoothness across nodes. Comprehensive experiments show the effectiveness of ADA-UGNN.

## 1 Introduction

Graph Neural Networks (GNNs) have shown great capacity in learning representations for graph-structured data and thus have facilitated many down-stream tasks such as node classification (Kipf & Welling, 2016; Veličković et al., 2017; Ying et al., 2018a; Klicpera et al., 2018) and graph classification (Defferrard et al., 2016; Ying et al., 2018b). As traditional deep learning models, a GNN model is usually composed of several stacking GNN layers. Given a graph $\mathcal{G}$ with $N$ nodes, a GNN layer typically contains a feature transformation and a feature aggregation operation as:

$$\text{Feature Transformation: } \mathbf{X}'_{in} = f_{trans}(\mathbf{X}_{in}); \quad \text{Feature Aggregation: } \mathbf{X}_{out} = f_{agg}(\mathbf{X}'_{in}; \mathcal{G}) \quad (1)$$

where $\mathbf{X}_{in} \in \mathbb{R}^{N \times d_{in}}$ and $\mathbf{X}_{out} \in \mathbb{R}^{N \times d_{out}}$ denote the input and output features of the GNN layer with $d_{in}$ and $d_{out}$ as the corresponding dimensions, respectively. Note that the non-linear activation is not included in Eq. (1) to ease the discussion. The feature transformation operation $f_{trans}(\cdot)$ transforms the input of $\mathbf{X}_{in}$ to $\mathbf{X}'_{in} \in \mathbb{R}^{N \times d_{out}}$ as its output; and the feature aggregation operation $f_{agg}(\cdot; \mathcal{G})$ updates the node features by aggregating the transformed node features via the graph $\mathcal{G}$.

In general, different GNN models share similar feature transformations (often, a single feed-forward layer), while adopting different designs for aggregation operation. We raise a natural question – is there an intrinsic connection among these feature aggregation operations and their assumptions? The significance of a positive answer to this question is two-fold. Firstly, it offers a new perspective to create a uniform understanding on representative aggregation operations. Secondly, it enables us to develop a general GNN framework that not only provides a unified view on multiple existing representative GNN models, but also has the potential to inspire new ones. In this paper, we aim to build the connection among feature aggregation operations of representative GNN models including GCN (Kipf & Welling, 2016), GAT (Veličković et al., 2017), PPNP and APPNP (Klicpera et al., 2018). In particular, we mathematically establish that the aggregation operations in these models can be unified as the process of exactly, and sometimes approximately, addressing a graph signal denoising problem with Laplacian regularization (Shuman et al., 2013). This connection suggests that these aggregation operations share a unified goal: to ensure feature smoothness of connected nodes. With this understanding, we propose a general GNN framework, UGNN, which not only provides a straightforward, unified view for many existing aggregation operations, but also suggests various promising directions to build new aggregation operations suitable for distinct applications. To demonstrate its potential, we build an instance of UGNN called ADA-UGNN, which is suited for handling varying smoothness properties across nodes, and conduct experiments to show its effectiveness.

## 2 REPRESENTATIVE GRAPH NEURAL NETWORKS

In this section, we introduce notations for graphs and briefly summarize several representative GNN models. A graph can be denoted as $\mathcal{G} = \{\mathcal{V}, \mathcal{E}\}$, where $\mathcal{V}$ and $\mathcal{E}$ are its corresponding node and edge sets. The connections in $\mathcal{G}$ can be represented as an adjacency matrix $\mathbf{A} \in \mathbb{R}^{N \times N}$, with $N$ the number of nodes in the graph. The Laplacian matrix of the graph $\mathcal{G}$ is denoted as $\mathbf{L}$. It is defined as $\mathbf{L} = \mathbf{D} - \mathbf{A}$, where $\mathbf{D}$ is a diagonal degree matrix corresponding to $\mathbf{A}$. There are also normalized versions of the Laplacian matrix such as $\mathbf{L} = \mathbf{I} - \mathbf{D}^{-\frac{1}{2}}\mathbf{A}\mathbf{D}^{-\frac{1}{2}}$ or $\mathbf{L} = \mathbf{I} - \mathbf{D}^{-1}\mathbf{A}$. In this work, we sometimes adopt different Laplacians to establish connections between different GNNs and the graph denoising problem, clarifying in the text. In this section, we generally use $\mathbf{X}_{in} \in \mathbb{R}^{N \times d_{in}}$ and $\mathbf{X}_{out} \in \mathbb{R}^{N \times d_{out}}$ to denote input and output features of GNN layers. Next, we describe a few representative GNN models.

### 2.1 GRAPH CONVOLUTIONAL NETWORKS (GCN)

Following Eq. (1), a single layer in GCN (Kipf & Welling, 2016) can be written as follows:

$$\text{Feature Transformation: } \mathbf{X}'_{in} = \mathbf{X}_{in}\mathbf{W}; \quad \text{Feature Aggregation: } \mathbf{X}_{out} = \tilde{\mathbf{A}}\mathbf{X}'_{in}, \tag{2}$$

where $\mathbf{W} \in \mathbb{R}^{d_{in} \times d_{out}}$ is a feature transformation matrix, and $\tilde{\mathbf{A}}$ is a normalized adjacency matrix which includes a self-loop, defined as follows:

$$\tilde{\mathbf{A}} = \hat{\mathbf{D}}^{-\frac{1}{2}}\hat{\mathbf{A}}\hat{\mathbf{D}}^{-\frac{1}{2}}, \quad \text{with} \quad \hat{\mathbf{A}} = \mathbf{A} + \mathbf{I} \quad \text{and} \quad \mathbf{D} = \text{diag}(\sum_j \hat{\mathbf{A}}_{1,j}, \ldots, \sum_j \hat{\mathbf{A}}_{N,j}). \tag{3}$$

In practice, multiple GCN layers can be stacked, where each layer takes the output of its previous layer as input. Non-linear activation functions are included between consecutive layers.

### 2.2 GRAPH ATTENTION NETWORKS (GAT)

Graph Attention Networks (GAT) adopts the same feature transformation operation as GCN in Eq. (2). The feature aggregation operation (written node-wise) for a node $i$ is as:

$$\mathbf{X}_{out}[i,:] = \sum_{j \in \tilde{\mathcal{N}}(i)} \alpha_{ij}\mathbf{X}'_{in}[j,:], \quad \text{with} \quad \alpha_{ij} = \frac{\exp(e_{ij})}{\sum_{k \in \tilde{\mathcal{N}}(i)} \exp(e_{ik})}. \tag{4}$$

where $\tilde{\mathcal{N}}(i) = \mathcal{N}(i) \cup \{i\}$ denotes the neighbors (self-inclusive) of node $i$, and $\mathbf{X}_{out}[i,:]$ is the $i$-th row of the matrix $\mathbf{X}_{out}$, i.e. the output node features of node $i$. In this aggregation operation, $\alpha_{ij}$ is a learnable attention score to differentiate the importance of distinct nodes in the neighborhood. Specifically, $\alpha_{ij}$ is a normalized form of $e_{ij}$, which is modeled as:

$$e_{ij} = \text{LeakyReLU}\left(\left[\mathbf{X}'_{in}[i,:]\|\mathbf{X}'_{in}[j,:]\right]\mathbf{a}\right) \tag{5}$$

where $[\cdot\|\cdot]$ denotes the concatenation operation and $\mathbf{a} \in \mathbb{R}^{2d}$ is a learnable vector. Similar to GCN, a GAT model usually consists of multiple stacked GAT layers.

### 2.3 PERSONALIZED PROPAGATION OF NEURAL PREDICTIONS (PPNP)

Personalized Propagation of Neural Predictions (PPNP) (Klicpera et al., 2018) introduces an aggregation operation based on Personalized PageRank (PPR). Specifically, the PPR matrix is defined as $\alpha(\mathbf{I} - (1-\alpha)\tilde{\mathbf{A}})^{-1}$, where $\alpha \in (0,1)$ is a hyper-parameter. The $ij$-th element of the PPR matrix specifies the influence of node $i$ on node $j$. The feature transformation operation is modeled as Multi-layer Perception (MLP). The PPNP model can be written in the form of Eq. (1) as follows:

$$\text{Feature Transformation: } \mathbf{X}'_{in} = \text{MLP}(\mathbf{X}_{in});$$

$$\text{Feature Aggregation: } \mathbf{X}_{out} = \alpha(\mathbf{I} - (1-\alpha)\tilde{\mathbf{A}})^{-1}\mathbf{X}'_{in}. \tag{6}$$

Unlike GCN and GAT, PPNP only consists of a single feature aggregation layer, but with a potentially deep feature transformation. Since the matrix inverse in Eq. (6) is costly, Klicpera et al. (2018) also introduces a practical, approximated version of PPNP, called APPNP, where the aggregation operation is performed in an iterative way as:

$$\mathbf{X}_{out}^{(k)} = (1-\alpha)\tilde{\mathbf{A}}\mathbf{X}_{out}^{(k-1)} + \alpha\mathbf{X}'_{in} \quad k = 1, \ldots K, \tag{7}$$

where $\mathbf{X}_{out}^{(0)} = \mathbf{X}'_{in}$ and $\mathbf{X}_{out}^{(K)}$ is the output of the feature aggregation operation. As proved in Klicpera et al. (2018), $\mathbf{X}_{out}^{(K)}$ converges to the solution obtained by PPNP, i.e., $\mathbf{X}_{out}$ in Eq. (6).

## 3 GNNS AS GRAPH SIGNAL DENOISING

In this section, we aim to establish the connections between the introduced GNN models and a graph signal denoising problem with Laplacian regularization. We first introduce the problem.

**Problem 1** (Graph Signal Denoising with Laplacian Regularization). *Suppose that we are given a noisy signal $\mathbf{X} \in \mathbb{R}^{N \times d}$ on a graph $\mathcal{G}$. The goal of the problem is to recover a clean signal $\mathbf{F} \in \mathbb{R}^{N \times d}$, assumed to be smooth over $\mathcal{G}$, by solving the following optimization problem:*

$$\arg\min_{\mathbf{F}} \quad \mathcal{L} = \|\mathbf{F} - \mathbf{X}\|_F^2 + c \cdot tr(\mathbf{F}^\top \mathbf{L} \mathbf{F}), \tag{8}$$

Note that the first term guides $\mathbf{F}$ to be close to $\mathbf{X}$, while the second term $tr(\mathbf{F}^\top \mathbf{L} \mathbf{F})$ is the Laplacian regularization that guides the smoothness of $\mathbf{F}$ over the graph. $c > 0$ is a balancing constant. Assuming we adopt the unnormalized version of Laplacian matrix with $\mathbf{L} = \mathbf{D} - \mathbf{A}$ (the adjacency matrix $\mathbf{A}$ is assumed to be binary), the second term in Eq. (8) can be written in an edge-centric way or a node-centric way as:

$$\text{edge-centric: } c \sum_{(i,j) \in \mathcal{E}} \|\mathbf{F}[i,:] - \mathbf{F}[j,:]\|_2^2; \quad \text{node-centric: } \frac{1}{2} c \sum_{i \in \mathcal{V}} \sum_{j \in \tilde{\mathcal{N}}(i)} \|\mathbf{F}[i,:] - \mathbf{F}[j,:]\|_2^2. \tag{9}$$

Clearly, from the edge-centric view, the regularization term measures the *global smoothness* of $\mathbf{F}$, which is small when connected nodes share similar features. On the other hand, we can view the term $\sum_{j \in \tilde{\mathcal{N}}(i)} \|\mathbf{F}[i,:] - \mathbf{F}[j,:]\|_2^2$ as a *local smoothness* measure for node $i$ as it measures the difference between node $i$ and all its neighbors. The regularization term can then be regarded as a summation of local smoothness over all nodes. Note that the adjacency matrix $\mathbf{A}$ is assumed to be binary when deriving Eq. (9). Similar formulations can also be derived to other types of Laplacian matrices. In the following subsections, we demonstrate the connections between aggregation operations in various GNN models and the graph signal denoising problem.

### 3.1 CONNECTION TO PPNP AND APPNP

In this subsection, we establish the connection between the graph signal denoising problem (8) and the aggregation propagations in PPNP and APPNP in Theorem 1 and Theorem 2, respectively.

**Theorem 1.** *When we adopt the normalized Laplacian matrix $\mathbf{L} = \mathbf{I} - \tilde{\mathbf{A}}$, with $\tilde{\mathbf{A}}$ defined in Eq. (3), the feature aggregation operation in PPNP (Eq. (6)) can be regarded as exactly solving the graph signal denoising problem (8) with $\mathbf{X}'_{in}$ as the input noisy signal and $c = \frac{1}{\alpha} - 1$.*

*Proof.* Note that the objective in Eq. (8) is convex. Hence, its closed-form solution $\mathbf{F}^*$ to exactly solve the graph signal denosing problem can be obtained by setting its derivative to $\mathbf{0}$ as:

$$\frac{\partial \mathcal{L}}{\partial \mathbf{F}} = 2(\mathbf{F} - \mathbf{X}) + 2c\mathbf{L}\mathbf{F} = 0 \Rightarrow \mathbf{F}^* = (\mathbf{I} + c\mathbf{L})^{-1} \mathbf{X} \tag{10}$$

Given $\mathbf{L} = \mathbf{I} - \tilde{\mathbf{A}}$, $\mathbf{F}^*$ can be reformulated as:

$$\mathbf{F}^* = (\mathbf{I} + c\mathbf{L})^{-1} \mathbf{X} = \left(\mathbf{I} + c\left(\mathbf{I} - \tilde{\mathbf{A}}\right)\right)^{-1} \mathbf{X} = \frac{1}{1+c} \left(\mathbf{I} - \frac{c}{1+c}\tilde{\mathbf{A}}\right)^{-1} \mathbf{X} \tag{11}$$

The feature aggregation operation in Eq. (6) is equivalent to the closed-form solution in Eq. (11) when we set $\alpha = 1/(1 + c)$ and $\mathbf{X} = \mathbf{X}'_{in}$. This completes the proof. $\square$

**Theorem 2.** *When we adopt the normalized Laplacian matrix $\mathbf{L} = \mathbf{I} - \tilde{\mathbf{A}}$, the feature aggregation operation in APPNP (Eq. (7)) approximately solves the graph signal denoising problem (8) by iterative gradient descent with $\mathbf{X}'_{in}$ as the input noisy signal, $c = \frac{1}{\alpha} - 1$ and stepsize $b = \frac{1}{2+2c}$.*

*Proof.* To solve the graph signal denoising problem (8), we take iterative gradient method with the stepsize $b$. Specifically, the $k$-th step gradient descent on problem (8) is as follows:

$$\mathbf{F}^{(k)} \leftarrow \mathbf{F}^{(k-1)} - b \cdot \frac{\partial \mathcal{L}}{\partial \mathbf{F}}(\mathbf{F} = \mathbf{F}^{(k-1)}) = (1 - 2b - 2bc)\mathbf{F}^{(k-1)} + 2b\mathbf{X} + 2bc\tilde{\mathbf{A}}\mathbf{F}^{(k-1)} \tag{12}$$

where $\mathbf{F}^{(0)} = \mathbf{X}$. When we set the stepsize $b$ as $\frac{1}{2+2c}$, we have the following iterative steps:

$$\mathbf{F}^{(k)} \leftarrow \frac{1}{1+c}\mathbf{X} + \frac{c}{1+c}\tilde{\mathbf{A}}\mathbf{F}^{(k-1)}, k = 1, \dots K, \tag{13}$$

which is equivalent to the iterative aggregation operation of the APPNP model in Eq. (7) with $\mathbf{X} = \mathbf{X}'_{in}$ and $\alpha = \frac{1}{1+c}$. This completes the proof. $\square$

These two connections provide a new explanation on the hyper-parameter $\alpha$ in PPNP and APPNP from the graph signal denoising perspective. Specifically, a smaller $\alpha$ indicates a larger $c$, which means the obtained $\mathbf{X}_{out}$ is enforced to be smoother over the graph.

### 3.2 CONNECTION TO GCN

We draw the connection between the GCN model (Kipf & Welling, 2016) and the graph signal denoising problem in Theorem 3.

**Theorem 3.** *When we adopt the normalized Laplacian matrix* $\mathbf{L} = \mathbf{I} - \tilde{\mathbf{A}}$*, the feature aggregation operation in GCN Eq. (2) can be regarded as solving the graph signal denoising problem (8) using one-step gradient descent with* $\mathbf{X}'_{in}$ *as the input noisy signal and stepsize* $b = \frac{1}{2c}$*.*

*Proof.* The gradient with respect to $\mathbf{F}$ at $\mathbf{X}$ is $\frac{\partial \mathcal{L}}{\partial \mathbf{F}}\big|_{\mathbf{F}=\mathbf{X}} = 2c\mathbf{L}\mathbf{X}$. Hence, one-step gradient descent for the graph signal denoising problem (8) can be described as:

$$\mathbf{F} \leftarrow \mathbf{X} - b\,\frac{\partial \mathcal{L}}{\partial \mathbf{F}}\bigg|_{\mathbf{F}=\mathbf{X}} = \mathbf{X} - 2bc\mathbf{L}\mathbf{X} = (1 - 2bc)\mathbf{X} + 2bc\tilde{\mathbf{A}}\mathbf{X}. \tag{14}$$

When stepsize $b = \frac{1}{2c}$ and $\mathbf{X} = \mathbf{X}'_{in}$, we have $\mathbf{F} \leftarrow \tilde{\mathbf{A}}\mathbf{X}'_{in}$, which is the same as the aggregation operation of GCN. □

With this connection, it is easy to verify that a GCN model with multiple GCN layers can be regarded as solving the graph signal denoising problem multiple times with different noisy signals. Specifically, each layer of a GCN model corresponds to a graph signal denoising problem, where the input noisy signal is the output from the previous layer after the feature transformation of the current layer. Note that there are earlier works (NT & Maehara, 2019; Zhao & Akoglu, 2019) drawing connection between GCN and the optimization problem in Eq. (8), where the aggregation operation in GCN is shown to be the first-order approximation of the exact solution.

## 3.3 CONNECTION TO GAT

To establish the connection between graph signal denoising and GAT (Veličković et al., 2017), in this subsection, we adopt an unnormalized version of the Laplacian. It is defined based on the adjacency matrix with self-loop $\hat{\mathbf{A}}$, i.e. $\mathbf{L} = \hat{\mathbf{D}} - \hat{\mathbf{A}}$ with $\hat{\mathbf{D}}$ denoting the diagonal degree matrix of $\hat{\mathbf{A}}$. Then, the denoising problem in Eq. (8) can be rewritten from a node-centric view as:

$$\arg\min_{\mathbf{F}} \ \mathcal{L} = \sum_{i\in\mathcal{V}} \|\mathbf{F}[i,:] - \mathbf{X}[i,:]\|_2^2 + \frac{1}{2}\sum_{i\in\mathcal{V}} c \cdot \sum_{j\in\tilde{\mathcal{N}}(i)} \|\mathbf{F}[i,:] - \mathbf{F}[j,:]\|_2^2, \tag{15}$$

where $\tilde{\mathcal{N}}(i) = \mathcal{N}(i) \cup \{i\}$ denotes the neighbors (self-inclusive) of node $i$. In Eq. (15), the constant $c$ is shared by all nodes, which indicates that the same level of *local smoothness* is enforced to all nodes. However, nodes in a real-world graph can have varied local smoothness. For nodes with low local smoothness, we should impose a relatively smaller $c$, while for those nodes with higher local smoothness, we need a larger $c$. Hence, instead of a unified $c$ as in Eq. (15), we could consider a node-dependent $c_i$ for each node $i$. Then, the optimization problem in Eq. (15) can be adjusted as:

$$\arg\min_{\mathbf{F}} \ \mathcal{L} = \sum_{i\in\mathcal{V}} \|\mathbf{F}[i,:] - \mathbf{X}[i,:]\|_2^2 + \frac{1}{2}\sum_{i\in\mathcal{V}} c_i \cdot \sum_{j\in\tilde{\mathcal{N}}(i)} \|\mathbf{F}[i,:] - \mathbf{F}[j,:]\|_2^2 \tag{16}$$

We next show that the aggregation operation in GAT is closely connected to an approximate solution of problem (16) with the help of the following theorem.

**Theorem 4.** *With adaptive stepsize* $b_i = 1/\sum_{j\in\tilde{\mathcal{N}}(i)}(c_i + c_j)$ *for each node* $i$*, the process of taking one step of gradient descent from* $\mathbf{X}$ *to solve problem (16) can be described as follows:*

$$\mathbf{F}[i,:] \leftarrow \sum_{j\in\tilde{\mathcal{N}}(i)} b_i(c_i + c_j)\mathbf{X}[j,:]. \tag{17}$$

*Proof.* The gradient of optimization problem in Eq. (16) with respect to $\mathbf{F}$ focusing on a node $i$ can be formulated as:

$$\frac{\partial \mathcal{L}}{\partial \mathbf{F}[i,:]} = 2\left(\mathbf{F}[i,:] - \mathbf{X}[i,:]\right) + \sum_{j\in\tilde{\mathcal{N}}(i)} (c_i + c_j)\left(\mathbf{F}[i,:] - \mathbf{F}[j,:]\right), \tag{18}$$

where $c_j$ in the second term appears since $i$ is also in the neighborhood of $j$. Then, the gradient at $\mathbf{X}$ is $\frac{\partial \mathcal{L}}{\partial \mathbf{F}[i,:]}\big|_{\mathbf{F}[i,:]=\mathbf{X}[i,:]} = \sum_{j\in\tilde{\mathcal{N}}(i)} (c_i + c_j)\left(\mathbf{X}[i,:] - \mathbf{X}[j,:]\right)$. Thus, taking a step of gradient descent starting from $\mathbf{X}$ with stepsize $b$ can be described as follows:

$$\mathbf{F}[i,:] \leftarrow \mathbf{X}[i,:] - b \cdot \frac{\partial \mathcal{L}}{\partial \mathbf{F}[i,:]}\bigg|_{\mathbf{F}[i,:]=\mathbf{X}[i,:]} = \left(1 - b\sum_{j\in\tilde{\mathcal{N}}(i)}(c_i + c_j)\right)\mathbf{X}[i,:] + \sum_{j\in\tilde{\mathcal{N}}(i)} b(c_i + c_j)\mathbf{X}[j,:] \tag{19}$$

Given $b = 1/\sum_{j\in\tilde{\mathcal{N}}(i)}(c_i + c_j)$, Eq. (19) can be rewritten as $\mathbf{F}[i,:] \leftarrow \sum_{j\in\tilde{\mathcal{N}}(i)} b_i(c_i + c_j)\mathbf{X}[j,:]$, which completes the proof. □

Eq. (17) resembles the aggregation operation of GAT in Eq. (4) if we treat $b_i(c_i + c_j)$ as the attention score $\alpha_{ij}$. Note that we have $\sum_{j \in \mathcal{N}(i)} (c_i + c_j) = 1/b_i$, for all $i \in \mathcal{V}$. So, $(c_i + c_j)$ can be regarded as the pre-normalized attention score and $1/b_i$ can be regarded as the normalization constant. We further compare $b_i(c_i + c_j)$ with $\alpha_{ij}$ by investigating the formulation of $e_{ij}$ in Eq. (5). Eq. (5) can be rewritten as:

$$e_{ij} = \text{LeakyReLU} \left( \mathbf{X}'_{in}[i, :]\mathbf{a}_1 + \mathbf{X}'_{in}[j, :]\mathbf{a}_2 \right) \tag{20}$$

where $\mathbf{a}_1 \in \mathbb{R}^d$ and $\mathbf{a}_2 \in \mathbb{R}^d$ are learnable column vectors, which can be concatenated to form $\mathbf{a}$ in Eq. (5). Comparing $e_{ij}$ with $(c_i + c_j)$, we find that they take a similar form. Specifically, $\mathbf{X}'_{in}[i, :]\mathbf{a}_1$ and $\mathbf{X}'_{in}[j, :]\mathbf{a}_2$ can be regarded as the approximations of $c_i$ and $c_j$, respectively. The difference between $b_i(c_i + c_j)$ and $\alpha_{ij}$ is that the normalization in Eq. (17) for $b_i(c_i + c_j)$ is achieved via summation rather than a softmax as in Eq. (4) for $\alpha_{ij}$. Note that since GAT makes the $c_i$ and $c_j$ learnable, they also include a non-linear activation in calculating $e_{ij}$. By viewing the attention mechanism in GAT from the perspective of Eq. (17), namely that $c_i$ actually indicates a notion of local smoothness for node $i$, we can develop other ways to parameterize $c_i$. For example, instead of directly using the node features of $i$ as an indicator of local smoothness like GAT, we can consider the neighborhood information. In fact, we adopt this idea to design a new aggregation operation in Section 5.

## 4 UGNN: A UNIFIED GNN FRAMEWORK VIA GRAPH SIGNAL DENOISING

In the previous section, we established that the aggregation operations in PPNP, APPNP, GCN and GAT are intimately connected to the graph signal denoising problem with (generalized) Laplacian regularization. In particular, from this perspective, all their aggregation operations aim to ensure feature smoothness: either a global smoothness over the graph as in PPNP, APPNP and GCN, or a local smoothness for each node as in GAT. This understanding allows us to develop a unified feature aggregation operation by posing the following, more general graph signal denoising problem:

**Problem 2** (Generalized UGNN Graph Signal Denoising Problem).

$$\arg\min_{\mathbf{F}} \quad \mathcal{L} = \|\mathbf{F} - \mathbf{X}\|_F^2 + r(\mathcal{C}, \mathbf{F}, \mathcal{G}), \tag{21}$$

*where $r(\mathcal{C}, \mathbf{F}, \mathcal{G})$ denotes a flexible regularization term to enforce some prior over $\mathbf{F}$.*

Note that we overload the notation $\mathcal{C}$ here: it can function as a scalar (like a global constant in GCN), a vector (like node-wise constants in GAT) or even a matrix (edge-wise constants) if we want to give flexibility to each node pair. Different choices of $r(\cdot)$ imply different feature aggregation operations. Besides PPNP, APPNP, GCN and GAT, there are aggregation operations in more GNN models that can be associated with Problem 2 with different regularization terms such as PairNorm (Zhao & Akoglu, 2019) and DropEdge (Rong et al., 2019) (more details can be found in Appendix B). The above mentioned regularization terms are all related to the Laplacian regularization. Other regularization terms can also be adopted, which may lead to novel designs of GNN layers. For example, if we aim to enforce that the clean signal is piece-wise linear, we can adopt $r(\mathcal{C}, \mathbf{F}, \mathcal{G}) = \mathcal{C} \cdot \|\mathbf{LF}\|_1$ designed for trend filtering (Tibshirani et al., 2014; Wang et al., 2016).

With these discussions, we propose a unified framework (UGNN) to design GNN layers from the graph signal processing perspective as: (1) Design a graph regularization term $r(\mathcal{C}, \mathbf{F}, \mathcal{G})$ in Problem 2 according to specific applications; (2) Feature Transformation: $\mathbf{X}'_{in} = f_{trans}(\mathbf{X}_{in})$; and (3) Feature Aggregation: Solving Problem 2 with $\mathbf{X} = \mathbf{X}'_{in}$ and the designed $r(\mathcal{C}, \mathbf{F}, \mathcal{G})$. To demonstrate the potential of UGNN, next we introduce a new GNN model ADA-UGNN by instantiating UGNN with $r(\mathcal{C}, \mathbf{F}, \mathcal{G})$ enforcing adaptive local smoothness across nodes. Note that we introduce ADA-UGNN with node classification as the downstream task.

## 5 ADA-UGNN: ADAPTIVE LOCAL SMOOTHING WITH UGNN

From the graph signal denoising perspective, PPNP, APPNP, and GCN enforces global smoothness by penalizing the difference with a constant $\mathcal{C}$ for all nodes. However, real-world graphs may consist of multiple groups of nodes which have different behaviors in connecting to similar neighbors. For example, Section 6.1 shows several graphs with varying distributions of local smoothness (as measured by label homophily): summarily, not all nodes are highly label-homophilic, and some nodes have considerably "noisier" neighborhoods than others. Moreover, as suggested by Wu et al. (2019); Jin et al. (2020), adversarial attacks on graphs tend to promote such label noise in graphs by connecting nodes from different classes and disconnecting nodes from the same class, rendering

resultant graphs with varying local smoothness across nodes. Under these scenarios, a constant $\mathcal{C}$ might not be optimal and adaptive (i.e. non-constant) smoothness to different nodes is desired. As shown in Section 3.3 by viewing GAT's aggregation as a solution to regularized graph signal denoising, GAT can be regarded as adopting an adaptive $\mathcal{C}$ for different nodes, which facilitates adaptive local smoothness. However, in GAT, the graph denoising problem is solved by a single step of gradient descent, which might still be suboptimal. Furthermore, when modeling the local smoothness factor $c_i$ in Eq. (17), GAT only uses features of node $i$ as input, which may not be optimal since by understanding $c_i$ as local smoothness, it should be intrinsically related to the *neighborhood* of node $i$. In this section, we adapt this notion directly into the UGNN framework by introducing a new regularization term, and develop a resulting GNN model (ADA-UGNN) which aims to enforce adaptive local smoothness to nodes in a different manner to GAT. We then utilize an iterative gradient descent method to approximate the optimal solution for Problem 2 with the following regularization term:

$$r(\mathcal{C}, \mathbf{F}, \mathcal{G}) = \frac{1}{2} \cdot \sum_{i \in \mathcal{V}} \mathcal{C}_i \sum_{j \in \tilde{\mathcal{N}}(i)} \left\| \frac{\mathbf{F}[i,:]}{\sqrt{d_i}} - \frac{\mathbf{F}[j,:]}{\sqrt{d_j}} \cdot \right\|_2^2 \tag{22}$$

where $d_i, d_j$ denotes the degree of node $i$ and $j$ respectively, and $\mathcal{C}_i$ indicates the smoothness factor of node $i$, which is assumed to be a fixed scalar. Note that, the above regularization term can be regarded as a generalized version of the regularization term used in PPNP, APPNP, and GCN. Similar to PPNP and APPNP, ADA-UGNN only consists of a single GNN layer. However, ADA-UGNN assumes adaptive local smoothness. We next describe the feature transformation and aggregation operations of ADA-UGNN, and show how to derive the model via UGNN.

## 5.1 FEATURE TRANSFORMATION

Similar to PPNP and APPNP, we adopt MLP for the feature transformation. Specifically, for a node classification task, the dimension of the output of the feature transformation $\mathbf{X}'_{in}$ is the number of classes in the graph.

## 5.2 FEATURE AGGREGATION

We use iterative gradient descent to solve Problem 2 with the regularization term in Eq. (22) The iterative gradient descent steps are stated in the following theorem and its proof can be found at Appendix A.1.

**Theorem 5.** *With adaptive stepsize* $b_i = 1 / \left( 2 + \sum\limits_{j \in \tilde{\mathcal{N}}(i)} (\mathcal{C}_i + \mathcal{C}_j)/d_i \right)$ *for each node $i$, the iterative gradient descent steps to solve Problem 2 with the regularization term in Eq. (22) is as follows:*

$$\mathbf{F}^{(k)}[i,:] \leftarrow 2b\mathbf{X}[i,:] + b_i \sum_{j \in \tilde{\mathcal{N}}(i)} (\mathcal{C}_i + \mathcal{C}_i) \frac{\mathbf{F}^{(k-1)}[j,:]}{\sqrt{d_i d_j}}; \quad k = 1, \ldots. \tag{23}$$

*where* $\mathbf{F}^{(0)}[i,:] = \mathbf{X}[i,:]$.

The iterative steps in Eq. (23) is guaranteed for convergence as stated in the following theorem and its proof can be found in Appendix A.2.

**Theorem 6.** *The iterative steps in Eq. (23) is guaranteed to converge to the optimal solution of Problem 2 with Eq. (22) as regularization term.*

Following the iterative solution in Eq. (23), we model the aggregation operation (for node $i$) for ADA-UGNN as follows:

$$\mathbf{X}_{out}^{(k)}[i,:] \leftarrow 2b_i \mathbf{X}'_{in}[i,:] + b_i \sum_{v_j \in \tilde{\mathcal{N}}(v_i)} (\mathcal{C}_i + \mathcal{C}_j) \frac{\mathbf{X}_{out}^{(k-1)}[j,:]}{\sqrt{d_i d_j}}; \quad k = 1, \ldots K, \tag{24}$$

where $K$ is the number gradient descent iterations, $\mathcal{C}_i$ can be considered as a positive scalar to control the level of "local smoothness" for node $i$ and $b_i$ can be calculated from $\{\mathcal{C}_j | j \in \tilde{\mathcal{N}}(i)\}$ as $b_i = 1 / \left( 2 + \sum\limits_{j \in \tilde{\mathcal{N}}(i)} (\mathcal{C}_i + \mathcal{C}_j)/d_i \right)$. However, in practice, $\mathcal{C}_i$ is usually unknown. One possible solution is to treat $\mathcal{C}_i$ as hyper-parameters. Treating $\mathcal{C}_i$ as hyper-parameters for all nodes is impractical, since there are, in total $N$ of them and we do not have their prior knowledge. Thus, we model $\mathcal{C}_i$ as a function of the information of the neighborhood of node $i$ as follows:

$$\mathcal{C}_i = s \cdot \sigma \left( h_1 \left( h_2 \left( \left\{ \mathbf{X}'_{in}[j,:] | j \in \tilde{\mathcal{N}}(i) \right\} \right) \right) \right), \tag{25}$$

where $h_2(\cdot)$ is a function to transform the neighborhood information of node $i$ to a vector, while $h_1(\cdot)$ further transforms it to a scalar. $\sigma(\cdot)$ denotes the sigmoid function, which maps the output scalar from $h_1(\cdot)$ to $(0, 1)$ and $s$ can be treated as a hyper-parameter controlling the upper bound of $\mathcal{C}_i$. $h_1(\cdot)$ can be modeled as a single layer fully-connected neural network. There are different designs for $h_2(\cdot)$ such as channel-wise variance or mean (Corso et al., 2020). In this paper, we adopt channel-wise variance as the $h_2(\cdot)$ function. In this case, the calculation of $\mathcal{C}_i$ in Eq. (25) only involves $H$ parameters, with $H$ denoting number of classes in the dataset. APPNP can be regarded a special case of ADA-UGNN, where $h_2(\cdot)$ is modeled as a constant function producing 1 as the output for all nodes. For the node classification task, the representation $\mathbf{X}_{out}^{(K)}$, which is obtained after $K$ iterations as in Eq. (24), is directly softmax normalized row-wise and its $i$-th row indicates the discrete class distribution of node $i$.

## 6 EXPERIMENT

In this section, we evaluate how the proposed ADA-UGNN handles graphs with varying local smoothness. We conduct node classification experiments on natural graphs, and also evaluate the model's robustness under adversarial attacks. We note that our main goal in proposing/evaluating ADA-UGNN is to demonstrate the promise of deriving new aggregations as solutions of denoising problems, rather than state-of-the-art performance.

### 6.1 NODE CLASSIFICATION

In this section, we conduct the node classification task. We first introduce the datasets and the experimental settings in Section 6.1.1 and then present the results in Section 6.1.2.

#### 6.1.1 DATASETS AND EXPERIMENTAL SETTINGS

We conduct the node classification task on 8 datasets from various domains including citation, social, co-authorship and co-purchase networks. Specifically, we use three citation networks including CORA, CITESEER, and PUBMED (Sen et al., 2008); one social network, BLOGCATALOG (Huang et al., 2017); two co-authorship networks including COAUTHOR-CS and COAUTHOR-PH (Shchur et al., 2018); and two co-purchase networks including AMAZON-COMP and Amazon Photos (Shchur et al., 2018). Descriptions and detail statistics about these datasets can be found in Appendix C.1. To provide a sense of the local smoothness properties of these datasets, in addition to the summary statistics, we also illustrate the *local label smoothness* distributions in Appendix C.1.1: here, we define the local label smoothness of a node as the ratio of nodes in its neighborhood that share the same label (see formal definition in Eq. (34) in Appendix C.1.1). Notably, the variety in local label smoothness within several real-world datasets – also observed in (Shah, 2020) – clearly motivates the importance of the adaptive smoothness assumption in ADA-UGNN. For the citation networks, we use the standard split as provided in Kipf & Welling (2016); Yang et al. (2016). For BLOG-CATALOG, we adopt the split provided in Zhao et al. (2020). For both the citation networks and BLOGCATALOG, the experiments are run with 30 random seeds and the average results are reported. For co-authorship and co-purchase networks, we utilize 20 labels per class for training, 30 nodes per class for validation and the remaining nodes for test. This process is repeated 20 times, which results in 20 different training/validation/test splits. For each split, the experiment is repeated for 20 times with different initialization. The average results over $20 \times 20$ experiments are reported. We compare our methods with the methods introduced in Section 2 including GCN, GAT and APPNP. Note that we do not include PPNP as it is difficult to scale for most of the datasets due to the calculation of inverse in Eq. 6. For all methods, we tune the hyperparameters from the following options: 1) learning rate: $\{0.005, 0.01, 0.05\}$; 2) weight decay $\{5e-04, 5e-05, 5e-06, 5e-07, 5e-08\}$; and 3) dropout rate: $\{0.2, 0.5, 0.8\}$. For APPNP and our method we further tune the number of iterations $K$ and the upper bound $s$ for $c_i$ in Eq. (25) from the following range: 1) $K$: $\{5, 10\}$; and $s$: $\{1, 9, 19\}$. Note that we treat APPNP as a special case of our proposed method with $h_2(\cdot) = 1$.

#### 6.1.2 PERFORMANCE COMPARISON

The performance comparison is shown in Table 1, where $t$-test is used to test the significance. First, GAT outperforms GCN in most datasets. It indicates that modeling adaptive local smoothness is helpful. Second, APPNP/ADA-UGNN outperform GCN/GAT in most settings, suggesting that iterative gradient descent may offer advantages to single-step gradients, due to their better ability to achieve a solution closer to the optimal. Third, and most notably, the proposed ADA-UGNN achieves consistently better performance than GCN/GAT, and outperforms or matches the state-of-the-art APPNP across datasets. Notice that in some datasets such as CORA, CITESEER, and

Table 1: Node Classification Accuracy on Various Datasets

| Dataset | GCN | GAT | APPNP | ADA-UGNN |
|---|---|---|---|---|
| CORA | 81.75±0.8 | 82.56±0.8 | 84.49±0.6 | 84.59±0.8* |
| CITESEER | 70.13±1.0 | 70.77±0.8 | 71.97±0.6 | 72.05±0.5 |
| PUBMED | 78.56±0.5 | 78.88±0.5 | 79.92±0.5 | 79.70±0.4 |
| BLOGCATALOG | 71.38±2.7 | 72.90±1.2 | 92.43±0.9 | 93.33±0.3*** |
| AMAZON-COMP | 82.79±1.3 | 83.01±1.5 | 82.99±1.6 | 83.40±1.3*** |
| AMAZON-PHOTO | 89.60±1.5 | 90.33±1.2 | 91.38±1.2 | 91.44±1.2 |
| COAUTHOR-CS | 91.55±0.6 | 90.95±0.7 | 91.69±0.4 | 92.33±0.5*** |
| COAUTHOR-PH | 93.23±0.7 | 92.86±0.7 | 93.84±0.5 | 93.92±0.6** |

$*, **, ***$ indicate the improvement over APPNP is significant at $p < 0.1, 0.05$ and $0.005$

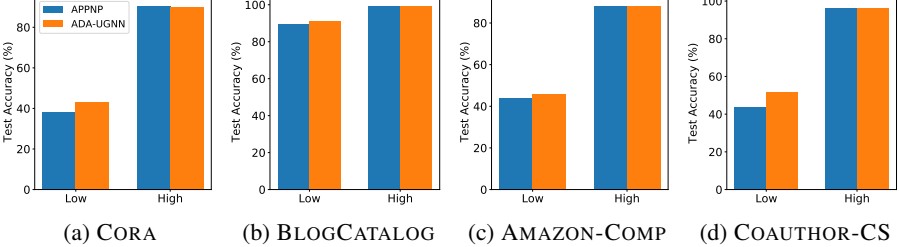

| (a) CORA | (b) BLOGCATALOG | (c) AMAZON-COMP | (d) COAUTHOR-CS |

Figure 1: Accuracy for nodes with low and high local label smoothness.

COAUTHOR-PH, the improvements of the proposed model compared with APPNP are not very significant. Figure 3 in Appendix C.1.1 shows that these datasets have extremely skewed local label smoothness distributions, with the majority of nodes having perfect, 1.0, label homophily (they are only connected to other nodes of the same label). APPNP shines in such cases, since its assumption of $h_2(\cdot) = 1$ is ideal for these nodes (designating maximal local smoothness). Conversely, our model has the challenging task of learning $h_2(\cdot)$ – in such skewed cases, learning $h_2(\cdot)$ may be quite challenging and unfruitful. On the other hand, for datasets with higher diversity in local label smoothness across nodes such as BLOGCATALOG and AMAZON-COMP, the proposed ADA-UGNN achieves more significant improvements.

To further validate, we partition the nodes in the test set of each dataset into two groups: (1) *high smoothness:* those with local label smoothness $>0.5$, and (2) *low smoothness:* those with $\leq 0.5$, and evaluate accuracy for APPNP and the proposed ADA-UGNN for each group. The results for CORA, BLOGCATALOG, AMAZON-COMP and COAUTHOR-CS are presented in Figure 1 while the results for the remaining datasets can be found in Figure 4 in Appendix C.2. Figure 1 clearly shows that ADA-UGNN consistently improves performance for low-smoothness nodes in most datasets, while keeping comparable (or marginally worse) performance for high-smoothness nodes. In cases where many nodes have low-level smoothness (like BLOGCATALOG or AMAZON-COMP), our method can notably improve overall performance.

## 6.2 ROBUSTNESS UNDER ADVERSARIAL ATTACKS

Adversarial attacks on graphs tend to connect nodes from different classes and remove edges between nodes from the same class (Wu et al., 2019; Jin et al., 2020), producing graphs with varying local label smoothness after attack (we demonstrate this in Appendix C.3). To further demonstrate that ADA-UGNN can handle graphs with varying local label smoothness better than alternatives, we conduct experiments to show its robustness under adversarial attacks. Specifically, we adopt Mettack (Zügner & Günnemann, 2019) to perform the attacks. Mettack produces non-targeted attacks which aim to impair test set node classification performance by strategically adding or removing edges from the victim graph. We utilize the attacked graphs (5%-25% perturb rate) from Jin et al. (2020) and follow the same setting, i.e., each method is run with 10 random seeds and the average performance is reported. These attacked graphs are generated from CORA, CITESEER and PUBMED, respectively and only the largest connected component is retained in each graph. Furthermore, the training, validation and test split ratio is $10/10/80\%$, which is different from the standard splits we use in Section 6.1. Thus, the performances reported in this section is not directly comparable with those in the previous section. We compare our method both with standard GNNs discussed in Section 2 (GCN, GAT, APPNP), but also with recent state-of-the-art defense techniques against adversarial attacks including GCN-Jaccard (Wu et al., 2019), GCN-SVD (Entezari et al., 2020), Pro-GNN-fs and Pro-GNN (Jin et al., 2020). The detailed description of these methods can be found at

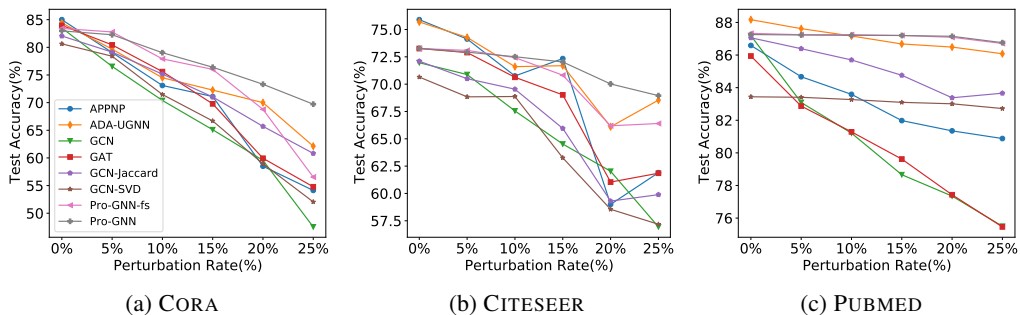

Figure 2: Robustness under adversarial attacks (node classification accuracy).

Appendix C.4. Results under varying perturbation rates (attack intensities) are shown in Figure 2. Again, we observe that GAT outperforms GCN, suggesting the appeal of an adaptive local smoothness assumption. Here, our method (orange) substantially outperforms GCN, GAT and APPNP by a large margin, especially in scenarios with high perturbation rate. Moreover, the proposed ADA-UGNN is also even more robust than several specially designed adversarial defense methods, like GCN-Jaccard and GCN-SVD, which are based on pre-processing the adversarial attack graphs to obtain cleaner ones, thanks to its adaptive smoothness assumption. Compared with Pro-GNN-fs, our method performs comparably or even better in a few settings, especially when perturbation rate is high. Furthermore, in these settings, the performance of our method is even closer to Pro-GNN, which is the current state-of-the art adversarial defense technique. Note that, Pro-GNN-fs and Pro-GNN involves learning cleaner adjacency matrices of the attacked graphs, and thus has $O(M)$ parameters (M denotes the number of edges in a graph), while our proposed model has far less parameters. Specifically, we have $O(d_{in} \cdot d_{out})$ for feature transformation and $H$ parameters for modelling $h_1(\cdot)$ with $H$ denoting the number of labels.

## 7 RELATED WORKS

There are mainly two streams of work in developing GNN models, i.e, spectral-based and spatial-based. When designing spectral-based GNNs, graph convolution (Shuman et al., 2013), defined based on spectral theory, is utilized to design graph neural network layers together with the feature transformation and non-linearity (Bruna et al., 2013; Henaff et al., 2015; Defferrard et al., 2016). These designs of the spectral-based graph convolution are tightly related with graph signal processing, and they can be regarded as graph filters. Low-pass graph filters can usually be adopted to denoise graph signals (Chen et al., 2014). In fact, most algorithms discussed in our work can be regarded as low-pass graph filters. With the emergence of GCN (Kipf & Welling, 2016), which can be regarded as a simplified spectral-based and also a spatial-based graph convolution operator, numerous spatial-based GNN models have since been developed (Hamilton et al., 2017; Veličković et al., 2017; Monti et al., 2017; Gao et al., 2018; Gilmer et al., 2017).

Graph signal denoising is to infer a cleaner graph signal given a noisy signal, and can be usually formulated as a graph regularized optimization problem (Chen et al., 2014). Recently, several works connect GCN with graph signal denoising with Laplacian regularization (NT & Maehara, 2019; Zhao & Akoglu, 2019), where they found the aggregation process in GCN models can be regarded as the first-order approximation of the optimal solution of the denoising problem. On the other hand, GNNs are also utilized to develop novel algorithms for graph denoising (Chen et al., 2020). Unlike these works, our paper details how a family of GNN models can be unified with a graph signal denoising perspective, and demonstrates its promise for new architecture design.

## 8 CONCLUSION

In this paper, we show how various representative GNN models including GCN, PPNP, APPNP and GAT can be unified mathematically as natural instances of graph denoising problems. Specifically, the aggregation operations in these models can be regarded as exactly or approximately addressing such denoising problems subject to Laplacian regularization. With these observations, we propose a general framework, UGNN, which enables the design of new GNN models from the denoising perspective via regularizer design. As an example demonstrating the promise of this paradigm, we instantiate the UGNN framework with a regularizer addressing adaptive local smoothness across nodes, a property prevalent in several real-world graphs, and proposed and evaluated a suitable new GNN model, ADA-UGNN.

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

# A  PROOFS

## A.1  PROOF OF THEOREM 5

**Theorem 5.** *With adaptive stepsize* $b_i = 1/\left(2 + \sum\limits_{v_j \in \tilde{\mathcal{N}}(v_i)} (\mathcal{C}_i + \mathcal{C}_j)/d_i\right)$ *for each node* $v_i$*, the iterative gradient descent steps to solve Problem 2 with the regularization term in Eq. (22) is as follows:*

$$\mathbf{F}^{(k)}[i,:] \leftarrow 2b\mathbf{X}[i,:] + b_i \sum_{v_j \in \tilde{\mathcal{N}}(v_i)} (\mathcal{C}_i + \mathcal{C}_i)\frac{\mathbf{F}^{(k-1)}[j,:]}{\sqrt{d_i d_j}}; \quad k = 1, \dots \tag{26}$$

*where* $\mathbf{F}^{(0)}[i,:] = \mathbf{X}[i,:]$*.*

*Proof.* The gradient of the optimization problem 2 with the regularization term in Eq. (22) with respect to $\mathbf{F}$ (focusing on node $i$) is as follows:

$$\frac{\partial \mathcal{L}}{\partial \mathbf{F}[i,:]} = 2(\mathbf{F}[i,:] - \mathbf{X}[i,:]) + \sum_{v_j \in \tilde{\mathcal{N}}(v_i)} \frac{\mathcal{C}_i + \mathcal{C}_j}{\sqrt{d_i}} \left(\frac{\mathbf{F}[i,:]}{\sqrt{d_i}} - \frac{\mathbf{F}[j,:]}{\sqrt{d_j}}\right), \tag{27}$$

where $\mathcal{C}_j$ in the second term appears since node $i$ is also in the neighborhood of node $j$. The iterative gradient descent steps with adaptive stepsize $b_i$ can be formulated as follows:

$$\mathbf{F}^{(k)}[i,:] \leftarrow \mathbf{F}^{(k-1)}[i,:] - b_i \cdot \left.\frac{\partial \mathcal{L}}{\partial \mathbf{F}[i,:]}\right|_{\mathbf{F}[i,:]=\mathbf{F}^{(k-1)}[i,:]}; \quad k = 1, \dots \tag{28}$$

With the gradient in Eq. (27), the iterative steps in Eq. (28) can be rewritten as:

$$\mathbf{F}^{(k)}[i,:] \leftarrow (1 - 2b_i - b_i \sum_{v_j \in \tilde{\mathcal{N}}(v_i)} \frac{\mathcal{C}_i + \mathcal{C}_j}{d_i})\mathbf{F}^{(k-1)}[i,:] + 2b_i\mathbf{X}[i,:]$$

$$+ b_i \sum_{v_j \in \tilde{\mathcal{N}}(v_i)} (\mathcal{C}_i + \mathcal{C}_j)\frac{\mathbf{F}^{(k)}[j,:]}{\sqrt{d_i d_j}}; \quad k = 1, \dots \tag{29}$$

Given $b_i = 1/\left(2 + \sum\limits_{v_j \in \tilde{\mathcal{N}}(v_i)} (\mathcal{C}_i + \mathcal{C}_j)/d_i\right)$, the iterative steps in Eq. (29) can be re-written as follows:

$$\mathbf{F}^{(k)}[i,:] \leftarrow 2b\mathbf{X}[i,:] + b_i \sum_{v_j \in \tilde{\mathcal{N}}(v_i)} (\mathcal{C}_i + \mathcal{C}_j)\frac{\mathbf{F}^{(k-1)}[j,:]}{\sqrt{d_i d_j}}; \quad k = 1, \dots, \tag{30}$$

with $\mathbf{F}^{(0)}[i,:] = \mathbf{X}[i,:]$, which completes the proof. $\qquad\square$

## A.2  PROOF OF THEOREM 6

**Theorem 6.** *The iterative steps in Eq. (23) is guaranteed to converge to the optimal solution of Problem 2 with Eq. (22) as regularization term.*

*Proof.* By taking the second derivative with respect to $\mathbf{F}[i,:]$, we obtain the Hessian matrix as:

$$\frac{\partial \mathcal{L}^2}{\partial \mathbf{F}[i,:]^2} = 2\mathbf{I} + \sum_{v_j \in \tilde{\mathcal{N}}(v_i)} (\frac{\mathcal{C}_i + \mathcal{C}_j}{d_i})\mathbf{I} \tag{31}$$

which implies the Lipschitz constant of the gradient in Eq. (27) is $2 + \sum\limits_{v_j \in \tilde{\mathcal{N}}(v_i)} (\frac{\mathcal{C}_i + \mathcal{C}_j}{d_i})$. To guarantee convergence, the stepsize $b_i$ for node $i$ should be smaller than $2/\left(2 + \sum\limits_{v_j \in \tilde{\mathcal{N}}(i)} (\mathcal{C}_i + \mathcal{C}_j)/d_i\right)$ (Nesterov, 2013). The stepsize we adopt in Theorem 5 is $b_i = 1/\left(2 + \sum\limits_{j \in \tilde{\mathcal{N}}(i)} (\mathcal{C}_i + \mathcal{C}_j)/d_i\right)$, hence the convergence is guaranteed. $\qquad\square$

## B Connections to PairNorm and DropEdge

PairNorm and DropEdge, which are two recently proposed GNN enhancements for developing deeper GNN models, are corresponding to the following regularization terms:

$$\text{PairNorm:} \quad \sum_{(i,j)\in\mathcal{E}} \mathcal{C}_p \cdot \|\mathbf{F}[i,:] - \mathbf{F}[j,:]\|_2^2 - \sum_{(i,j)\notin\mathcal{E}} \mathcal{C}_n \cdot \|\mathbf{F}[i,:] - \mathbf{F}[j,:]\|_2^2, \tag{32}$$

$$\text{DropEdge:} \quad \sum_{(i,j)\in\mathcal{E}} \mathcal{C}_{ij} \cdot \|\mathbf{F}[i,:] - \mathbf{F}[j,:]\|_2^2, \quad \text{where } \mathcal{C}_{ij} \in \{0,1\}. \tag{33}$$

For PairNorm, $\mathcal{C}$ consists of $\mathcal{C}_p, \mathcal{C}_n > 0$ and the regularization term ensures connected nodes to be similar while disconnected nodes to be dissimilar. For DropEdge, $\mathcal{C}$ is a sparse matrix having the same shape as adjacency matrix. For each edge $(i,j)$, its corresponding $\mathcal{C}_{ij}$ is sampled from a Bernoulli distribution with mean $1-q$, where $q$ is a pre-defined dropout rate.

## C Experiments

### C.1 Datasets

|  | #Nodes | #Edges | #Labels | #Features |
|---|---|---|---|---|
| Cora | 2708 | 13264 | 7 | 1433 |
| Citeseer | 3327 | 12431 | 6 | 3703 |
| Pubmed | 19717 | 108365 | 3 | 500 |
| BlogCatalog | 5196 | 348682 | 6 | 8189 |
| Amazon-Comp | 13381 | 504937 | 10 | 767 |
| Amazon-Photo | 7487 | 245573 | 8 | 745 |
| Coauthor-CS | 18333 | 182121 | 15 | 6805 |
| Coauthor-PH | 34493 | 530417 | 5 | 8415 |

Table 2: Dataset summary statistics.

In this section, we provide information of the datasets we used in the experiments as follows:

- **Citation Networks:** Cora, Citeseer and Pubmed are widely adopted benchmarks of GNN models. In these graphs, nodes represent documents and edges denote the citation links between them. Each node is associated bag-of-words features of its corresponding document and also a label indicating the research field of the document.

- **Blogcatalog:** BlogCatalog is an online blogging community where bloggers can follow each other. The BlogCatalog graph consists of blogger as nodes while their social relations as edges. Each blogger is associated with some features generated from key words of his/her blogs. The bloggers are labeled according to their interests.

- **Co-purchase Graph:** Amazon-Comp and Amazon-Photo are co-purchase graphs, where nodes represent items and edges indicate that two items are frequently bought together. Each item is associated with bag-of-words features extract from its corresponding reviews. The labels of items are given by the category of them.

- **Co-authorship Graphs:** Coauthor-CS and Coauthor-PH are co-authorship graphs, where nodes are authors and edges indicating the co-authorship between authors. Each author is associated with some features representing the keywords of his/her papers. The label of an author indicates the his/her most active research field.

Some statistics of these graphs are shown in Table 2.

#### C.1.1 Local Label Smoothness of Datasets

We further present the distribution of local label smoothness in these datasets. For a node $v_i$ we formally define the local label smoothness as follows

$$\text{ls}(i) = \frac{\sum_{j\in\mathcal{N}(i)} \mathbf{1}\{l(i) = l(j)\}}{|\mathcal{N}(i)|} \tag{34}$$

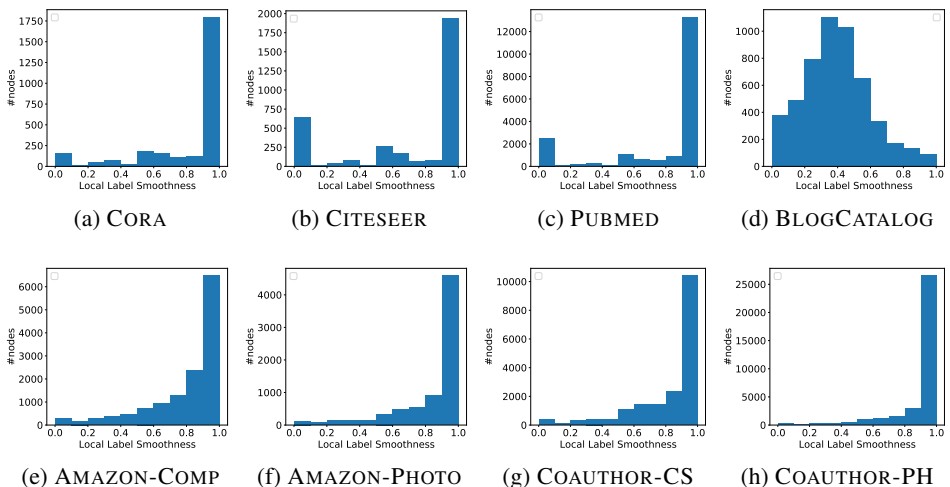

Figure 3: Distribution of local label smoothness (homophily) on different graph datasets: note the non-homogeneity of smoothness values.

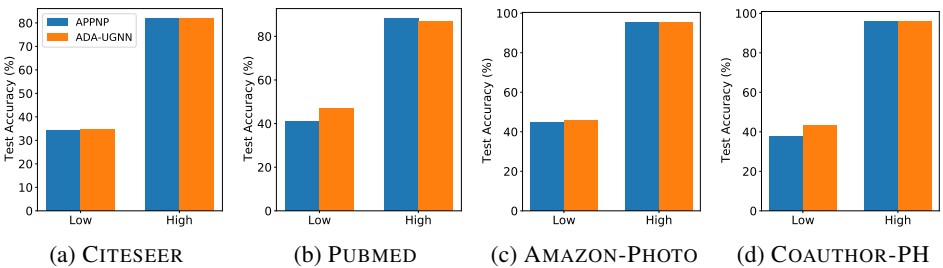

Figure 4: Accuracy with low label smoothness and high label smoothness nodes. Note the consistent improvement in low smoothness cases, enabled by adaptive local smoothing.

where $l(v_i)$ denotes the label of node $v_i$ and $\mathbf{1}\{a\}$ is an indicator function, which takes 1 as output only when $a$ is true, otherwise 0. The distributions of local label smoothness for all 8 datasets are presented in Figure 3.

### C.2 NODE CLASSIFICATION ACCURACY FOR NODES WITH LOW-LEVEL AND HIGH-LEVEL LOCAL LABEL SMOOTHNESS

The performance of nodes with low local label smoothness and high local label smoothness in CITE-SEER, PUBMED, AMAZON-PHOTO and COAUTHOR-PH are presented in Figure 4.

### C.3 LOCAL SMOOTHNESS DISTRIBUTION OF ATTACKED GRAPH

Graph adversarial attacks tend to connect nodes from different classes while disconnect nodes from the same class, which typically leads to more diverse distributions of local smoothness level. We present the distributions of the graphs generated by Mettack (Zügner & Günnemann, 2019) with different perturbation rate for CORA, CITESEER and PUBMED in Figure 5, Figure 6 and Figure 7, respectively.

### C.4 BASELINES FOR ADVERSARIAL DEFENSE

In this section, we list the descriptions of the defense algorithms we adopt in Section 6.2 as follows:

- **GCN-Jaccard (Wu et al., 2019):** GCN-Jaccard aims to pre-process a given attacked graph by removing those edges added by the attackers. Specifically, Jaccard smilarlity is utilized

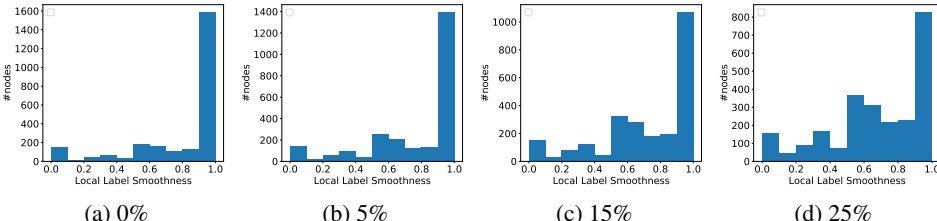

Figure 5: Distribution of local label smoothness on CORA with various attack perturbation rates.

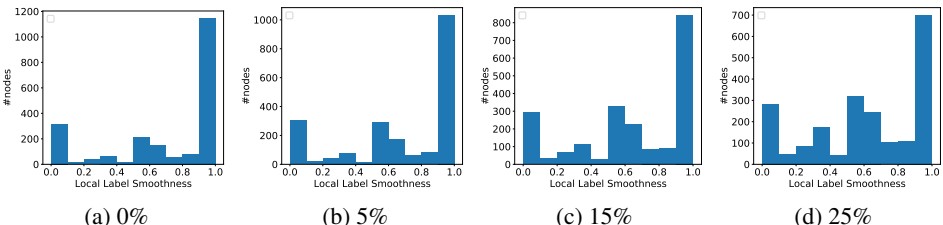

Figure 6: Distribution of local label smoothness on CITESEER with various attack perturbation rates.

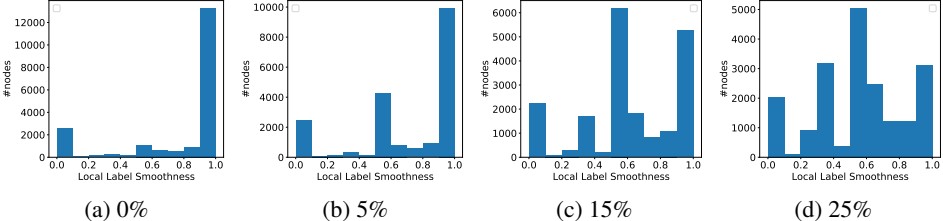

Figure 7: Distribution of local label smoothness on PUBMED with various attack perturbation rates.

> to measure the feature similarity between connected pairs of nodes. The edges between node pairs with low-similarity are removed by the algorithm. This pre-processed graph is then utilized for the node classification task.

- **GCN-SVD (Entezari et al., 2020):** GCN-SVD is also a pre-process method. It use SVD to decompose the adjacency matrix of a given perturbed graph and then obtain its low-rank approximation. The low-rank approximation is believed to be cleaner as graph adversarial attacks are observed to be high-rank in (Entezari et al., 2020).

- **Pro-GNN (Jin et al., 2020):** Pro-GNN tries to learn a cleaner graph while training the node classification model at the same time. Specifically, it treats the adjacency as parameters, which is optimized during the training stage. Several different constraints are enforced to this learnable adjacency matrix, including: 1) the learned adjacency matrix should be close to the original adjacency matrix; 2) the learned adjacency matrix should be low-rank; and 3) the learned adjacency matrix should ensure feature smoothness. **Pro-GNN-fs** is a variant of Pro-GNN where the third constraint, i.e. feature smoothness, is not enforced.

## C.5 INVESTIGATION ON NUMBER OF GRADIENT DESCENT STEPS IN ADA-UGNN

In this section, we conducted experiments to check how the performance of ADA-UGNN is affected by $K$. For each $K$, we run the experiments on standard splits of CORA, CITESEER and PUBMED with 30 random seeds (i.e., the same setting as in Section 6.) The average performance is reported. As shown in Figure 8, the performance increases quickly as $K$ gets larger when $K$ is relatively small. After $K$ becomes large, the performance either slowly grows or slightly fluctuates as $K$ further increases.

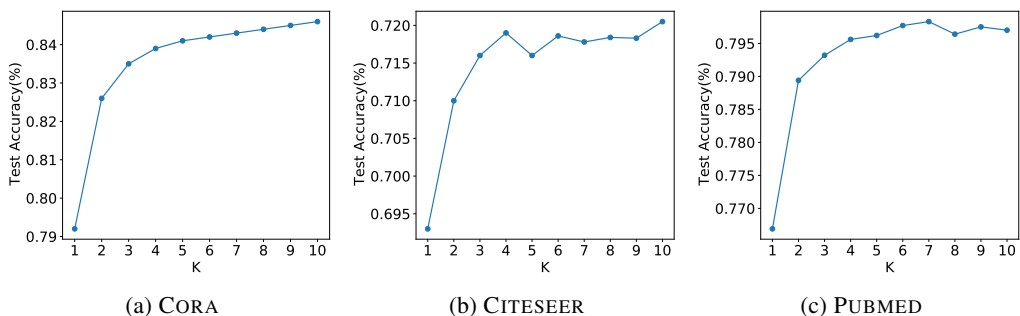

(a) CORA        (b) CITESEER        (c) PUBMED

Figure 8: ADA-UGNN performance (test accuracy) under different numbers of gradient steps ($K$).

