# OpenReview forum: "A Unified View on Graph Neural Networks as Graph Signal Denoising"
_ICLR.cc/2021/Conference — Reject_

### Official Review · AnonReviewer1 · 2020-10-26
**A Unified View on Graph Neural Networks as Graph Signal Denoising**

**Rating:** 7
**Confidence:** 4

**Review:**

The paper attempts to provide a unified picture of a number of different GNN architectures from the point of view of graph signal processing (GSP).
In particular, it is argued that the aggregation operation in a number of important architectures can effectively be seen as a form of (graph) signal denoising.

Pros:
* Consolidates a number of (a priori) unrelated GNN architectures and unifies them from the point of view of GSP.
* The unified picture can be used for systematic comparison of architectures or for the design of new architectures, as the authors show.

Cons:
* It appears that despite the theoretical insights are nice for unifying the representation, but do not quite lead to a more improved GNN architecture in terms of numerical performance.
* The paper concentrates only on the aggregation step, but in the architectures considered there are also different non-linearities -- this part is completely neglected.

More detailed comments.

Overall I very much appreciate the work towards simplifying and unifying the somewhat disconnected literature -- this is an good contribution, I think.
While some of the transformations the authors propose to make the architectures align with the GSP picture feel a bit contrived, I think that is ok, given the simplification gained.
What I am missing somewhat was a discussion in how far knowing this representation can actually theoretically lead to improvements. For instance it is known that GNNs cannot be more expressive that the weisfeiler-leman test (see Morris et al) --- does the GSP lens provide any alternative perspective on this? This would be very interesting theoretically I think.

My second suggestion for possible improvements would be to provide a somewhat more detailed discussion on the effects of the non-linearity in such networks for the classification results.
Unifying the aggregation layer is useful, but at the end the improvements that these insights bring (in terms of UGNN) seem to somewhat small.

Overall I see the strongest contribution here in terms of the theoretical unification provided.

Minor comment:
The Laplacian is invariant to the addition of self loops, i.e., \hat{D} - \hat{A} = D-A, as the addition of a self loop on the diagonal of A will be cancelled by the corresponding change in D.
(cf. top of page 4)

---

> ### Author Response · Authors · 2020-11-18
> **Response to AnonReviewer1**
>
> We highly appreciate your positive comments. Your valuable questions/suggestions also inspired us to further think the proposed framework. Our responses to the comments/suggestions are as follows.
> ***
> Q1. What I am missing somewhat was a discussion in how far knowing this representation can actually theoretically lead to improvements. For instance, it is known that GNNs cannot be more expressive than the Weisfeiler-Leman test (see Morris et al) --- does the GSP lens provide any alternative perspective on this? This would be very interesting theoretically I think.
>
> A1. We think the graph signal denoising provides a complementary perspective on the theoretical work regarding of the expressivity of GNNs.  Summarily, although past expressivity works and our work both offer theoretical understanding regarding of GNNs, the former focuses on a structure-first perspective, while ours focuses on a features-first perspective.
>
> The Weisfeiler-Lehman test (WL-test)
> primarily focused on structural similarity, being a test for graph isomorphism.  In this case, node features are designated to be "identifiers'' for the nodes to distinguish themselves from each other. GNNs are shown to be as most as powerful as WL-test in capturing the graph structural information [1].  On the other hand, the graph signal denoising perspective discussed in this work particularly focuses more on the node features, and the aggregation process over graph structure as a means of denoising these node features under smoothness assumption on the underlying graph. This smoothness assumption is implicitly used by a few existing GNN models (including the ones we discussed in this paper); here, the graph structure information is used mainly in the specification of the regularization term to denoise the given node features. This term can be understood as enforcing a prior to the node features into the representation learning process. For example, the Laplacian regularization term we discussed in Problem 1 corresponds to a prior that the expected graph signal (node features) is smooth over the graph (or roughly "connected nodes tend to share similar features"). Hence, GNN models corresponding to the Laplacian regularization term such as GCN and APPNP can work well on datasets (such as CORA, CITESEER, and PUBMED) where node features adhere well to this prior. We conjecture that the above mentioned GNN models are likely to not work well on datasets that break this smoothness assumption (or prior knowledge).
> Thus, we believe that our framework provides alternative understandings of intrinsic smoothness assumptions of GNN models and a culprit as to why they don't work well in certain data settings. Also, it offers a perspective on developing novel GNN models by carefully designing the regularization terms according to prior knowledge of specific tasks/datasets.
>
> [1] Xu, K., Hu, W., Leskovec, J., \& Jegelka, S. "How Powerful are Graph Neural Networks? '', ICLR'2019.
> ***
> Q2. Provide a somewhat more detailed discussion on the effects of the non-linearity in such networks for the classification results.
>
> A2. We conducted experiments to check how the non-linearity affects the performance of ADA-UGNN framework. Specifically, we run additional experiments for the ADA-UGNN framework following exactly the same setting as in the experiments section. The results
>  (with activation, without activation) are listed in the following table.
>
> |      | CORA  | CITESEER | PUBMED | BLOGCATALOG | AMAZON-COMP | AMAZON-PHOTO | COAUTHOR-CS | COAUTHOR-PH |
> |------|-------|----------|--------|-------------|-------------|--------------|-------------|-------------|
> | with | 84.59 | 72.05    | 79.70  | 93.33       | 83.40       | 91.44        | 92.33       | 93.92       |
> | w/o  | 84.03 | 71.86    | 70.85  | 93.33       | 83.03       | 91.62        | 92.33       | 93.80       |
>
> In most of the datasets, introducing the non-linearity helps improve the performance. On some datasets, such as AMAZON-PHOTO and PUBMED, non-linear activation actually leads to a tiny performance drop. Overall, we found that the non-linearity does not impact the classification performance in a very significant way for the settings we considered. We leave the theoretical exploration of this effect as one future work.
> ***
> Q3. The Laplacian is invariant to the addition of self loops, i.e., $\hat{D} - \hat{A} = D-A$, as the addition of a self loop on the diagonal of A will be cancelled by the corresponding change in D. (cf. top of page 4)
>
> A3. Thanks for pointing this out. Yes, the self-loop will be canceled. We keep the self-loop to keep consistent with GAT since GAT's aggregation operation includes the node itself as its neighbors.

---

### Official Review · AnonReviewer3 · 2020-10-28
**The submission claimed to provide an connection between GCN,GAT,PPPN and APPNP using a denoising perspective. However, the contribution is limited and the results are not sufficiently strong.**

**Rating:** 3
**Confidence:** 4

**Review:**

1). The novelty and contribution are very limited. In literature, many papers have discussed the connection between different GNNs, typically, including aggregators and Updaters, such as discussed in “Deep Learning on Graphs: A Survey”. The submission only provides a kind of connection between GCN, GAT, PPPN and APPNP in the perspective of denoising. Compared with that, the survey paper actually connects many different GNNs.


2). The writing quality is low. There are many errors, for example, in section 3, “the unnormalized version of Laplacian matrix with L−D−A”==> “the unnormalized version of Laplacian matrix with L=D−A”?


3). In Eq. (1) to ease the discussion, the non-linear activation is not included. However, the nonlinearity is the key part for deep GNNs. In Formula (8), the GNNS AS GRAPH SIGNAL DENOISING actually oversimplified the topological smoothing over attributes, since the nonlinear transformation, especially with dropout will already conduct the signal denoting. So why use such additional effects to do that?

4). The submission provides both Node classification task and ADVERSARIAL defense task to validate the performance. As for Node classification, the results however are not very promising compared with current SOTA. For example, in ICLR’20 paper “ADAPTIVE STRUCTURAL FINGERPRINTS FOR GRAPH ATTENTION NETWORKS”, the cora dataset reports “85.4±0.3%” compared with that reported by this submission only “84.59±0.8”; in cite seer “74.0±0.4%” compared with this submission’s report “72.05±0.5”. In Pubmed, this paper reports “79.70±0.4”, however, in the ICLR paper, they report “81.2±0.3%”. From that perspective, I did not see any advantage in the submission.

5). As another task for validation, that is robustness to ADVERSARIAL attack. It is suggest to compare the recent SOTA “Graph Information Bottleneck” by Jure stanford in NeurIPS’20.

---

> ### Author Response · Authors · 2020-11-18
> **Response to AnonReviewer3 Part 1**
>
> On the one hand, we thank you for the volunteered time in reviewing our work. On the other hand, to us, your comments include some unreasonable misunderstandings of our work, if we understand your comments correctly. Thus, we appreciate it if you can engage in the discussion with us to clarify these misunderstandings from both sides. We believe that both your and our time devoted to this paper deserve thorough discussions.
> ***
> Q1. The novelty and contribution are very limited. In literature, many papers have discussed the connection between different GNNs, typically, including aggregators and Updaters, such as discussed in “Deep Learning on Graphs: A Survey”. The submission only provides a kind of connection between GCN, GAT, PPPN and APPNP in the perspective of denoising. Compared with that, the survey paper actually connects many different GNNs.
>
> A1. When you mention the connection among different GNNs from the survey “Deep Learning on Graphs: A Survey”, we assume that you mean the survey paper's citation of the MPNN framework [1]. Note that MPNN is mainly a very high-level framework for GNN layers: One of the key components for MPNN is the general message passing process as described in Eq. (1) and (2) in [1]. Note that the feature aggregation operation in our paper corresponds to the summation in Eq.~(1) in [1], and the feature transformation operation in our paper corresponds to the message functions $M_t()$ in Eq.(1) in [1]. We do not have a specific operation in our framework corresponding to Eq.(2) in [1]. Note that our novelty and contributions do not lie on the decomposition of GNN models into "feature aggregation" and "feature transformation" operations, which is commonly known in the literature; rather, we introduce these in the text to give readers a consistent understanding of GNN models.
>
> The novelty and contributions in our work are that we provide a unified view for the feature aggregation operations (or message passing process) for many GNN models: specifically, we dive deep into these operations. This is a novel contribution and has not been sufficiently explored in prior work. This is also quite different from both the mentioned survey, as well as the MPNN framework. We believe our findings in this paper can deepen the understandings of GNN models, as well as suggest the potential for developing new GNN models.
>
> [1] Gilmer, Justin, et al. "Neural message passing for quantum chemistry." arXiv preprint arXiv:1704.01212 (2017).
> ***
> Q2.  The writing quality is low. There are many errors, for example, in section 3, ''the unnormalized version of Laplacian matrix with L-D-A'' ''the unnormalized version of Laplacian matrix with L=D-A''?
>
> A2. This was a typo. We have revised this typo and also the entire paper with careful proofread.

---

> > ### Author Response · Authors · 2020-11-18
> > **Response to AnonReviewer3 Part 2**
> >
> > Q3. In Eq. (1) to ease the discussion, the non-linear activation is not included. However, nonlinearity is the key part for deep GNNs. In Formula (8), the GNNS AS GRAPH SIGNAL DENOISING actually oversimplified the topological smoothing over attributes, since the nonlinear transformation, especially with dropout will already conduct the signal denoting. So why use such additional effects to do that?
> >
> > A3. In this paper, we analyzed the key component of GNN models, i.e. the feature aggregation operation. Note that the nonlinear activation can be regarded as a function applied to the output of the feature transformation and feature aggregation operations. We did not include the non-linearity to keep the formulation concise. Actually, the non-linearity (and similarly the dropout) can be included in the proposed framework: Specifically, for models like PPNP and APPNP, which only have a single feature aggregation operation (also a single feature transformation operation), these can be incorporated into the feature transformation operation (i.e. MLP). Hence, our framework can naturally describe their entire models including the non-linearity and dropout (absorbed in the MLP for feature transformation).
> > On the other hand, GCN and GAT models usually consist of multiple layers. In this case, non-linearities can be regarded as additional operations between GNN layers. As mentioned in Section 3.2, "a GCN model with multiple GCN layers can be regarded as solving the graph signal denoising problem multiple times with different noisy signals. Specifically, each layer of a GCN model corresponds to a graph signal denoising problem, where the input noisy signal is the output from the previous layer after the feature transformation of the current layer." In these multi-layer cases, the non-linearity will have effects for the entire multiple-layer GCN models; we leave this analysis as one future work.
> >
> > Regarding the second point, we do not agree that "the nonlinear transformation, especially with dropout will already conduct the signal denoting''. The non-linearity generally does increase the model capacity of GNNs, and dropout is typically used to prevent over-fitting; however, these do not effectively conduct graph signal denoising. In fact, both non-linearities and dropout do not use the graph structure at all, much less conduct graph signal denoising; they are only related to the feature transformation, rather than the aggregation.  Additionally, we are not "using additional effects to" include the feature aggregation component, or introduce the graph signal denoising to GNN models. Instead, the significance of our work is to understand the feature aggregations of many GNNs as the graph denoising problem. Feature aggregations (or graph denoising) in many GNNs models have been proven to be helpful in boosting the performance. Thus, this further supports that your claim "the nonlinear transformation, especially with dropout will already conduct the signal denoting" is not valid.
> > ***
> > Q4 The submission provides both Node classification task and ADVERSARIAL defense task to validate the performance. As for Node classification, the results however are not very promising compared with current SOTA. ICLR’20 paper “ADAPTIVE STRUCTURAL FINGERPRINTS FOR GRAPH ATTENTION NETWORKS” achieves better performance than our method.
> >
> > A4. The major contribution of our paper is the presented a unified view for feature aggregation processes in different models. As claimed at the beginning of Section 6, "our main goal in proposing/evaluating ADA-UGNN is to demonstrate the promise of deriving new feature aggregation operations as solutions of the denoising problems, rather than state-of-the-art performance." In the experiments, we primarily compare ADA-UGNN with APPNP, which can be viewed as a special case of ADA-UGNN with constant $\mathcal{C}$. Note that ADA-UGNN is derived from a modified version of the graph denoising problem corresponding to APPNP. The major difference between ADA-UGNN and APPNP is that in ADA-UGNN, we adopt learnable, adaptive $\mathcal{C}$ as per the assumptions of the regularization term introduced in Eq. (22). The experiments demonstrate that making $\mathcal{C}$ adaptive and learnable can actually help improve the performance over APPNP in some of the datasets and especially under the setting of adversarial attack, where the local label smoothness distribution are largely perturbed. This experimental analysis is mainly to show that the proposed UGNN framework is capable of helping design GNN models better suited to certain data assumptions. The mentioned paper “ADAPTIVE STRUCTURAL FINGERPRINTS FOR GRAPH ATTENTION NETWORKS” aims to incorporate graph structure information to enhance the attention mechanism into GAT, which is out of the scope of our paper's objective.

---

> > > ### Author Response · Authors · 2020-11-18
> > > **Response to AnonReviewer3 Part 3**
> > >
> > > Q5. As another task for validation, that is robustness to ADVERSARIAL attack. It is suggest to compare the recent SOTA “Graph Information Bottleneck” by Jure stanford in NeurIPS’20.
> > >
> > > A5. Note that this paper, "Graph Information Bottleneck" was released on arXiv on October 24, 2020.  As the ICLR submission deadline for this work was October 2, 2020, it is unreasonable to expect a comparison with it. Furthermore, our intention for the experiments under adversarial attack is not to show that our method achieves state-of-the-art performance in defending adversarial attacks (this is not a stated goal of the work). Our goal with the relevant experiments is to use the attack methods to generate graphs with varying local label smoothness as a byproduct, which demonstrates that the proposed ADA-UGNN, which learns adaptive local smoothness score $\mathcal{C}_i$ for each node $i$, can achieve better performance than other GNN models by leveraging the correct regularization assumption.

---

### Official Review · AnonReviewer4 · 2020-10-29
**A Unified View on Graph Neural Networks as Graph Signal Denoising**

**Rating:** 6
**Confidence:** 3

**Review:**

Summary of the paper: In this paper, the authors make the following new argument: The aggregation processes of current popular GNN models such as GCN, GAT, PPNP, and APPNP can be treated as a graph denoising problem where the objective is to minimize a recovery error (a norm of noisy feature matrix, i.e. ||F-X||) plus a graph-based regularization (smoothness). This new view provides a way to build a GNN model, namely (Ada-)UGNN. Experimental results show the effectiveness of Ada-UGNN on the task of node classification and the task of preventing adversarial attacks on graphs.

Strong points: 1) Theoretical contributions of the proposed framework are solid and interesting. These findings show that two basic operations of a GNN layer, feature transformation and feature aggregation can be viewed as a gradient descent step of minimizing a graph denoising function. 2) Experimental results demonstrate the effectiveness of Ada-UGNN.

Weak points: 1) I think one weakness of this paper is that: Explanations are only focused on one layer (local). The theorems do not explain the relations between layers and how nonlinear activation functions affect these theoretical findings. For example, [1] and [2] treat the GNN as a procedure of encoding and decoding as a whole. However, it seems that the objective of GNN cannot be viewed as a simple combination of graph denoising problems. 2) The experiments do not explain well of theoretical findings: these connections are missing in experiments. I do see results of Ada-UGNN are promising on node classification and the task of preventing adversarial attacks. However, it would be better if there are some empirical evidence to explain these new theorems.

Recommendation: Based on the above points, I tend to marginally accept this paper but have concerns (these weak points).

Questions & other comments:
“The improvements of the proposed model compared with APPNP are marginal”, as shown in Table 1. Are these really improvements? Based on my understanding, these means of Ada-UGNN are higher than APPNP, but the variance is also high. Significance test is needed.
In Ada-UGNN, it approximately solves problem 2 and uses a special regularization term. How does the approximation affect the final performance? Is there any clear guidance on how to choose the regularization term in different problems? Are these regularizations problem-dependent?

[1] Hamilton, William L., Rex Ying, and Jure Leskovec. "Representation learning on graphs: Methods and applications." arXiv preprint arXiv:1709.05584 (2017).
[2] Chami, I., Abu-El-Haija, S., Perozzi, B., Ré, C., & Murphy, K. (2020). Machine Learning on Graphs: A Model and Comprehensive Taxonomy. arXiv preprint arXiv:2005.03675.

---

> ### Author Response · Authors · 2020-11-18
> **Response to AnonReviewer4 Part 1**
>
> Thanks for the positive comments and valuable questions. These questions have allowed us to improve the paper. Our response are as follows.
> ***
> Q1. I think one weakness of this paper is that: Explanations are only focused on one layer (local). The theorems do not explain the relations between layers and how nonlinear activation functions affect these theoretical findings. For example, [1] and [2] treat the GNN as a procedure of encoding and decoding as a whole. However, it seems that the objective of GNN cannot be viewed as a simple combination of graph denoising problems.
>
> A1.  In this paper, we analyzed the key component of GNN models, i.e. the feature aggregation operation. Note that the nonlinear activation can be regarded as a function applied to the output of the feature transformation and feature aggregation operations. Hence, in general, nonlinear activation does not influence the theoretical understanding provided in this paper (i.e., for the feature aggregation operation in a single GNN layer). APPNP and PPNP only have a single feature transformation operation and feature aggregation operation, so they can be naturally described by our framework by a single graph denoising problem.  On the other hand, GCN and GAT usually consist of multiple layers. As mentioned in Section 3.2, "a GCN model with multiple GCN layers can be regarded as solving the graph signal denoising problem multiple times with different noisy signals. Specifically, each layer of a GCN model corresponds to a graph signal denoising problem, where the input noisy signal is the output from the previous layer after the feature transformation of the current layer." We further illustrated this process in more detail as follows. Given $X$ as the input node features, the learning process of a $L$-layer GCN model is described in terms of pseudo code as follows:
>      $\quad$1.  **Input**: Node features $X$; Adjacency matrix $A$.
>      $\quad$2.  **Initialize** $X_{(0)}\leftarrow X$ ; $l\leftarrow 1$.
>      $\quad$3.  While ($1 \leq l\leq L$) do:
>          $\quad$$\quad$3.a (**Feature Transformation**) $X^f_{(l-1)} = X_{(l-1)}W_{(l-1)}$; $W_{(l-1)}$ is a transformation matrix to be learned.
>          $\quad$$\quad$3.b (**Feature Aggregation**) Let $X^f_{(l-1)}$ be the input noisy signal of Problem 1. Obtain the solution of Problem 1 by one-step gradient descent as described in Theorem 1 and denote the obtained solution as $X^g_{(l)}$.
>          $\quad$$\quad$3.c (**Activation**) $X_{(l)} = \sigma(X^g_{(l)})$, where we use $\sigma()$ to denote an activation function. $l \leftarrow l+1$.
>      $\quad$4.  **Output:** $ X_{(L)}$.
> According to the above description, the GCN model (more accurately the feature learning part -- the encoder according to [1] and [2](more discussions later) -- of the GCN model) can be viewed as a combination of a series of graph signal denoising problems.  Note that GAT models with multiple layers can be understood similarly.  In these multi-layer cases, the non-linearity will have effects for the entire multiple-layer GCN model; we leave this analysis for future work.
>
> [1] and [2] provide a very high-level framework for GNN models in terms of encoder-decoder. The encoder is usually used to learn the node representations (i.e. feature learning) and the decoder is to reconstruct (or infer) some kind of information from these learned node representations. From this encoder-decoder perspective, in this paper, we are investigating the encoders for different GNN models. More specifically, our main contribution is that we provide a unified view for the encoders in many different GNN models. Note that [1] and [2] do not build such connections between encoders from different GNN models. Also, from the encoder-decoder perspective, the encoder of GCN (or GAT) can be regarded as a combination of graph signal denoising problems as we discussed above.
>
> [1] Hamilton, William L., Rex Ying, and Jure Leskovec. "Representation learning on graphs: Methods and applications." arXiv preprint arXiv:1709.05584 (2017).
> [2] Chami, Ines, et al. "Machine Learning on Graphs: A Model and Comprehensive Taxonomy." arXiv preprint arXiv:2005.03675 (2020).

---

> > ### Author Response · Authors · 2020-11-18
> > **Response to AnonReviewer4 Part 2**
> >
> > Q2. The experiments do not explain well of theoretical findings: these connections are missing in experiments. I do see results of Ada-UGNN are promising on node classification and the task of preventing adversarial attacks. However, it would be better if there are some empirical evidence to explain these new theorems.
> >
> > A2. We conducted experiments to empirically investigate why ADA-UGNN is promising on node classification and the task of preventing adversarial attacks. Specifically, we investigate the correlation between the learned $\mathcal{C}$ scores and the local label smoothness (measured from ground truth labels). The local label smoothness of a node is defined as the ratio of nodes in its neighborhood that share the same label with the node (see formal definition in Eq. (34) in Appendix C.1.1). Ideally, for nodes with high local label smoothness, we expect the learned $\mathcal{C}$ to be larger, such that a higher-level local smoothness is enforced to this node. So, the learned $\mathcal{C}$ score is expected to be positively correlated with the local label smoothness. We adopt the Pearson correlation coefficient to measure the correlations. To understand how $\mathcal{C}$ affects the model in normal node classification and under the adversarial attack setting, we mostly investigate the models in Section 6.2, where $0%$ perturbation ratio is corresponding to the original clean graph.  The correlations on CORA, CITESEER, and PUBMED under different ratios of adversarial perturbations are listed as follows
> >
> > | Perturbation ratio | 0%      | 5%     | 10%    | 15%    | 20%    | 25%   |
> > |--------------------|---------|--------|--------|--------|--------|-------|
> > | Cora               |  0.415  | 0.457  | 0.487  | 0.471  | 0.459  | 0.448 |
> > | Citeseer           | 0.357   | 0.391  | 0.423  | 0.437  | 0.447  | 0.470 |
> > | Pubmed             | 0.426   | 0.569  | 0.600  | 0.615  | 0.606  | 0.587 |
> >
> > In general, the learned scores are substantially positively correlated with the local label smoothness under all settings on all three datasets. Furthermore, compared with the clean graph ($0%$ perturbation), the correlation scores are generally higher when the graphs are perturbed. This is likely because these three datasets all have very skewed local label smoothness distribution (a very large portion of the nodes has local label smoothness $1$) as discussed in Section 6.1.2 (the distributions can be found in Appendix C.1.1). Under perturbation, the local label smoothness distributions of these three datasets become much more diverse (See the distributions in Appendix C.3), which makes the process of learning $\mathcal{C}$ easier. These findings are consistent with our original conjecture in Section 6.1.2.  This partially explains why ADA-UGNN does not outperform APPNP (note that APPNP can be viewed as having constant $\mathcal{C}$ for all nodes) significantly on these three datasets under $0\%$ perturbation but can outperform by large margins under the attack setting. Note that on other datasets with diverse local label smoothness distribution, the learned $\mathcal{C}$ is also highly correlated with the local label smoothness. For example, on BLOGCATALOG and AMAZON-COMP, the correlation scores are 0.638 and 0.598, respectively. In general, we believe these empirical results demonstrate that our model can learn reasonable $\mathcal{C}$ to adaptively enforce appropriate levels of smoothness to different nodes, which help improve the model performances.
> > ***
> > Q3. The improvements of the proposed model compared with APPNP are marginal”, as shown in Table 1. Are these really improvements? Significance test is needed.
> >
> > A3. We conducted a two-sample t-test to test whether ADA-UGNN's improvement over APPNP is significant -- the results are included in Table 1 in the revised submission.  In detail, among the three datasets mentioned in the sentence "Notice that in some datasets such as CORA, CITESEER, and COAUTHOR-PH, the improvements of the proposed model compared with APPNP are marginal", the improvements on datasets CORA and COAUTHOR-PH are significant at $p<0.1$ and $p<0.05$, respectively. On CITESEER, the improvement is not significant.  We have modified the corresponding text in the paper from "marginal" to "not very significant".  Note that, we also originally provided reasons on why this phenomenon happens for these datasets in Section 6.1.2 of the paper -- specifically, these datasets all have very skewed local label smoothness distribution. Here, we define the local label smoothness of a node as the ratio of nodes in its neighborhood that share the same label (see formal definition in Eq. (34) in Appendix C.1.1. [in the updated version of the paper]) On the other hand, the improvements on datasets that have much more diverse local smoothness distribution like BLOGCATALOG and AMAZON-COMP are very significant ($p<0.005$).   Overall, ADA-UGNN outperforms or performs comparably to APPNP across the datasets.

---

> > > ### Author Response · Authors · 2020-11-18
> > > **Response to AnonReviewer4 Part 3**
> > >
> > > Q4. In Ada-UGNN, it approximately solves problem 2 and uses a special regularization term. How does the approximation affect the final performance?
> > >
> > > A4. We empirically investigate how the approximation affects the final performance. Note that, as described in Eq. (23) and Theorem 5, we approximately solve Problem 2 by iterative gradient descents ($K$ iterations). Hence, ideally, as the number of gradient descent iterations $K$ increases, the solution is expected to be more accurate, and thus the performance is expected to increase. We conducted experiments to check how the performance of ADA-UGNN is affected by $K$. Detailed results on CORA, CITSEER, and PUBMED can be found in Figure 8 in Appendix C.5. In summary, the results show that model performance increases sharply at small values of $K$, upon which we observe diminishing returns and stability/minor fluctuation as $K$ becomes larger. This shows that more iterations generally provide us with a more accurate solution for Problem 2, which results in better model performance. Furthermore, given enough iterations (i.e., large $K$), it converges to reasonable model performance. We leave the theoretical investigation as one future work.
> > > ***
> > > Q5. Is there any clear guidance on how to choose the regularization term in different problems? Are these regularizations problem-dependent?
> > >
> > > A5. Yes, these regularization terms can be problem-dependent. We can actually choose the regularization term based on prior knowledge about the given data. For example, the Laplacian regularization term is corresponding to the prior knowledge that the ground truth graph signal (node labels) is smooth over the graph (or, "connected nodes tend to have the same label or similar features").  However, if we believe that the node labels are piece-wise constant
> > > (i.e., there are some abrupt changes in the margin of clusters with different labels; this is actually quite common in many graph datasets), then, according to [3], graph trend filtering, which adopts $L_1$ based regularization terms, can adapt to this situation better than Laplacian regularization. Hence, an $L_1$-based regularization term such as $\sum\limits_{i\in \mathcal{V}} \sum\limits_{j\in \tilde{\mathcal{N}}(i)} \|{\bf F}[i,:]-{\bf F}[j,:] \|_1 $ and $\|{\bf L}{\bf F}\|_1$ might be more suitable for node classification task under this setting, and we can use them to design a potentially better GNN model, which is suitable for these datasets with this kind of prior knowledge.
> > > Another example is PairNorm [4], which is proposed to solve the oversmoothing issue in GNN models. PairNorm is based on the prior knowledge that "connected nodes tend to share similar features while disconnected nodes are likely to have different features." As described in Section 4 (detailed in Appendix B) of our paper, PairNorm is covered by our framework corresponding to the following regularization term $R_A+ R_B$ with $R_A$ and $R_B$ as follows:
> > >
> > > $R_A = \\sum\\limits_{(i,j) \\in \\mathcal{E}} c_p \\cdot \\left\\| {\bf F}[i,:] - {\\bf F}[j,:]\\right\\|_2^2 $ and  $R_B = - \\sum\\limits_{(i,j)\\not\\in \\mathcal{E}} c_n \\cdot \\| {\\bf F}[i,:] - {\\bf F}[j,:]\\|_2^2.$
> > >  This regularization term enforces the aforementioned prior knowledge. The above examples demonstrate how we can choose different regularization terms based on different prior knowledge about the dataset.
> > >
> > > [3] Wang, Yu-Xiang, et al. "Trend Filtering on Graphs." The Journal of Machine Learning Research 17.1 (2016): 3651-3691.
> > > [4] Zhao, Lingxiao, and Leman Akoglu.
> > >  ``PairNorm: Tackling Oversmoothing in GNNs'', ICLR'2020.

---

### Official Review · AnonReviewer2 · 2020-10-30
**Worth pursing idea but needs more work**

**Rating:** 3
**Confidence:** 5

**Review:**

SUMMARY:
This paper establishes a relation between different popular graph neural networks by mathematically proving that the feature aggregation operation of such networks can be understood as a graph-signal-denoising step. Moreover, the authors try to establish a general framework based on graph signal denoising that subsumes the studied architectures, developing new graph neural network (GNN) architecture under this framework.

STRONG POINTS:
Showing that different architectures are indeed using a similar approach for the feature aggregation which is closely related to graph signal denoising is an interesting idea which helps to gain insight into how these architectures work. Furthermore, the numerical results illustrate that the proposed architecture has a competitive performance under certain settings.

WEAK POINTS:
The mathematical notation of the paper is sometimes ambiguous and unclear, so it should be carefully revised.

The relation between GAT and the graph signal denoising approach is not clear and should be detailed, since it is one of the main contributions of the paper.

While proposing a unified GNN framework based on graph signal denoising is stated as one of the main contributions of the paper, it amounts to presenting the graph denoising formulation with an arbitrary regularization function. The paper should focus more on the relation between the proposed architecture and the different “GNN with graph signal denoising schemes”.
The paper would benefit if the proofs of Theorems 1-4 were included in the main body, rather than in the appendix.
The reason is twofold. Those proofs are likely to constitute the main contribution of the paper. Furthermore, the statement of the theorems (without the proofs) is not sufficient to fully illustrate the relation with graph signal denoising.

The proposed ADA-UGNN network should be further analyzed. An MLP is chosen as the feature aggregation function without providing a motivation. Furthermore, the impact of C being learned instead of being a hyperparameter (Theorems 5 and 6) should be discussed in more detailed, since it implies that convexity of (14) is lost.

ADDITIONAL COMMENTS:

The symbol L is ambiguously used to denote different types of Laplacian matrices.
In Section 2 says that each node is associated with a d-dimensional signal X of size N times d, but the signal associated with a node should be a vector, not a matrix.

Equation (4) is not mathematically correct. The variables used as indexes of the summation are not present in the terms inside the summation. In fact, notation for the indexes of the summations throughout the entire manuscript is quite confusing (and in cases like (4) definitely incorrect).

The edge-centric interpretation of the Laplacian regularization is never used. Also, both the edge-centric and node-centric formulations are not correct since they are missing the related term of the adjacency matrix A_ij. If A is binary, this should be stated clearly.

Two different notations for the gradient descent algorithms are used in the paper (see, e.g., equations (21) and (24) vs. (25) and (30)). This should be unified.

In eq (15), please clarify what d_i and d_j represent.

---

> ### Author Response · Authors · 2020-11-18
> **Response to AnonReviewer2 Part1**
>
> We appreciate the reviewer's detailed suggestions on the notation and organization of the paper; we have largely updated our paper according to the review. We also appreciate the questions from the reviewer. Detailed responses are as follows.
> ***
> Q1. The mathematical notation of the paper is sometimes ambiguous and unclear, so it should be carefully revised.
>
> A1. Thanks for your suggestions; we have carefully revised the notations throughout the entire paper. Details on these revisions can be found in the responses to later questions.
> ***
> Q2. The relation between GAT and the graph signal denoising approach is not clear and should be detailed.
>
> A2. The feature aggregation process in GAT is as follows:
> $$X_{out}[i,:] = \sum\limits_{i\in\tilde{\mathcal{N}}(i)} X_{in}[j,:], \quad \text{with} \quad \alpha_{ij}=\frac{\exp \left( e_{ij}\right)}{\sum\limits_{k \in \tilde{\mathcal{N}}(i)} \exp \left(e_{ik}\right)}. \quad(1)$$
> We found that this formulation is closely related to an approximate solution to the problem in Eq. (16) in the paper. According to Theorem 4, if we take one step of gradient with step size $b_i =1/\sum\limits_{j\in \tilde{\mathcal{N}}(i)}(c_i+c_j)$, we can obtain the following solution:
> \begin{align}
>     {\bf F}[i,:] \leftarrow   \sum\limits_{j\in \tilde{\mathcal{N}}(i)} b_i(c_i+c_j) {\bf X}[j,:],\quad (2)
> \end{align}
> which resembles the feature aggregation of GAT, if we treat $b_i(c_i +c_j)$ as the attention score, $\alpha_{ij}$ in Eq.(1). Note that, ${\bf F}$ and ${\bf X}$ in Eq. (2) are corresponding to $X_{out}$ and $X_{in}$ in Eq.(1), respectively. We have $\sum\limits_{j\in \tilde{\mathcal{N}}(i)}(c_i+c_j)=1/b_i$, for all $i\in \mathcal{V}$. So, $(c_i+c_j)$ can be regarded as the pre-normalized attention score and $1/b_i$ can be regarded as the normalization constant. More specifically, the connections between $b_i(c_i +c_j)$ and $\alpha_{ij}$ are detailed as follows.
> The pre-normalized attention score $e_{ij}$ in Eq.(1) can be calculated as  $e_{ij} = \text{LeakyReLU} (X_{in}[i,:]a_1 +  X_{in}[j,:]a_2)$, where $a_1$ and $a_2$ are parameters (column vectors) to be learned. Compared $e_{ij}$ with $(c_i+c_j)$, we can find they take a similar form. Specifically, $X_{in}[i,:]a_1$ and $X_{in}[j,:]a_2$ can be viewed as the approximations of $c_i$ and $c_j$, respectively. The difference between $b_i(c_i+c_j)$ and $\alpha_{ij}$ is that the normalization in Eq.(2) for $b_i(c_i+c_j)$ is achieved via summation rather than a softmax as in Eq.(1) for $\alpha_{ij}$. Note that since GAT makes the $c_i$ and $c_j$ learnable, they also include a non-linear activation in calculating $e_{ij}$. We have adjusted the corresponding explanations in the paper accordingly. We also moved the proof of Theorem 4 to the main text of the paper.
> ***
> Q3. The paper should focus more on the relation between the proposed architecture and the different “GNN with graph signal denoising schemes”. The paper would benefit if the proofs of Theorems 1-4 were included in the main body, rather than in the appendix.
>
> A3. We agree that our work's main contributions center around the mathematical unification of multiple widely used GNN methods under the regularized graph signal denoising perspective.  We adjusted the organization to include the proofs of Theorem 1-4 in the main text of the paper.
> ***
> Q4. An MLP is chosen as the feature aggregation function without providing motivation.
>
> A4. Note that we do not use MLP for feature aggregation, but rather for feature transformation (we think this might be a typo in the review). We use MLP accordingly for feature transformation since most GNN models use the same: either single-layer ones as in GCN and GAT, or multi-layer ones as in APPNP and PPNP. Hence, we follow this convention.

---

> > ### Author Response · Authors · 2020-11-18
> > **Response to AnonReviewer2 Part2**
> >
> > Q5. The impact of C being learned instead of being a hyperparameter (Theorems 5 and 6) should be discussed in more detailed, since it implies that convexity of (14) is lost.
> >
> > A5. We want to clarify that Theorem 5 and Theorem 6 are general, and not specific for the ADA-UGNN framework. They are derived under the condition that $\mathcal{C}_i$ is a fixed constant for node $i$. We have revised the text to made this point clear in the paper (See text under Eq.(22) in the updated paper). We use Theorem 5 to motivate the formulation of ADA-UGNN; specifically, we used the update steps derived in Theorem 5 to design the feature aggregation process in ADA-UGNN. However, since $\mathcal{C}_i$ is generally unavailable, we opt to treat them as parameters of the model. We avoid treating $\mathcal{C}_i$ as hyperparameters, because these would scale according to O(N), or the number of nodes in the graph. Hence, to make the feature aggregation more practical, we use a function to approximate $\mathcal{C}_i$. This function only introduces $d$ parameters, with $d$ indicating the dimension of node features.
> >
> > We empirically investigated the convergence (with respect to $K$) of the process in Eq. (23) in the updated paper (i.e., the gradient descent iterations) with learnable $\mathcal{C}$. We illustrate the performance of the ADA-UGNN (under the same experiment setting as in the original paper) with different number of gradient descent iterations $K$; we added detailed results on CORA, CITESEER, and PUBMED in Figure 8 in Appendix C.5. In summary, the results show that model performance increases sharply at small values of $K$, upon which we observe diminishing returns and stability/minor fluctuation as $K$ becomes larger.  This shows that even with learnable $\mathcal{C}$, we are still able to improve the solution iteration by iteration. Furthermore, given enough iterations (i.e., large $K$), it can converge to an empirically reasonable solution, which results in reasonable model performance. We leave the theoretical investigation as one future work.
> > ***
> > Q6. The symbol L is ambiguously used to denote different types of Laplacian matrices.
> >
> > A6. Different GNN models use different definitions of Laplacian matrices; hence, for convenience, we abuse notation so that $L$ can flexibly encompass these, in our general formulation.  We specified which Laplacian variant is used each time when we reference the Laplacian matrix specifically.
> > ***
> > Q7. In Section 2 says that each node is associated with a d-dimensional signal X of size N times d, but the signal associated with a node should be a vector, not a matrix.
> >
> > A7. This is a typo, and we have revised it throughout the paper.
> > ***
> > Q8. Equation (4) is not mathematically correct. The variables used as indexes of the summation are not present in the terms inside the summation. In fact, notation for the indexes of the summations throughout the entire manuscript is quite confusing (and in cases like (4) definitely incorrect).
> >
> > A8.  We have updated the paper to use $i$ (instead of $v_i$) to denote a single node. Now, the indexes should be correct.
> > ***
> > Q9. The edge-centric interpretation of the Laplacian regularization is never used. Also, both the edge-centric and node-centric formulations are not correct since they are missing the related term of the adjacency matrix $A_{ij}$. If A is binary, this should be stated clearly.
> >
> > A9. Thanks for your detailed comments on the edge-centric and node-centric formulations.  We use the edge-centric interpretation to provide an understanding of the Laplacian regularization term from a global perspective.  Furthermore, it is used in Section 4 to show how our framework can also describe PairNorm and DropEdge (details can be found in Appendix B). We were assuming a binary adjacency matrix when deriving these formulations. We adjusted the paper to make this point clear in the text, following the suggestion.
> > ***
> > Q10. Two different notations for the gradient descent algorithms are used in the paper (see, e.g., equations (21) and (24) vs. (25) and (30)). This should be unified. In eq (15), please clarify what $d_i$ and $d_j$ represent.
> >
> > A10. Thanks for your detailed comments; we have modified the paper accordingly. We made the notations for gradient descent consistent. We also added explanations for $d_i$ and $d_j$ (They are degrees of node i and j.).

---

### Official Review · AnonReviewer5 · 2020-11-07
**Nice insights, but unclear novelty and impact**

**Rating:** 6
**Confidence:** 3

**Review:**

**Post-discussion update:**

I would like to thank the authors for addressing (albeit partially) my comments, as well as the comments from other reviewers. While I understand that some connections can be made between the proposed approach and other approaches or aspects that go beyond local smoothing or oversmoothing, this is somewhat anecdotal in my opinion. More generally, it is still not entirely clear to me how significant are the contributions here. Reading the other reviews, it seems these concerns are also shared by other reviewers, although I still think the analysis here is not without merit and this warrants it at least a borderline score. Further, there are some interesting insights provided here, which place this work slightly over the threshold. Since marginally above the threshold was already the score I gave the manuscript initially, it remains unchanged.

---

**Original review:**

This work provides a formal interpretation of the aggregation step in common GNN architectures as aiming to solve (at least partially) a graph denoising problem formulated via graph Laplacian smoothing. In particular, it provides a unified formulation of GCN, GAT, PPNP and APPNP in these terms. Further, the authors use this understanding to formulate a unified GNN architecture, which can be instantiated to these particular ones, and then leverage this architecture to provide an improved adaptive one that is demonstrated to have some advantages over the mentioned classic GNNs.

The main insight showing that traditional GNN architectures essentially smooth or denoise node features is not surprising, and one might argue not quite novel, since this has been discussed and demonstrated in several previous work (some important ones already acknowledged and cited here). However, the unified view provided here does seem insightful and can contribute to more methodical view of their architecture design. Indeed, this is demonstrated by the introduction of ADA-UGNN here. Therefore, the paper does have merit that progresses the theory behind graph neural networks.

On the other hand, it is not clear how much insight can be drawn from the presented theory, or what impact this analysis might have. As said before, it is generally well accepted and understood that the four architectures discussed here rely on local smoothing, and are susceptible (to some degree) to oversmoothing in node classification. There is little discussion of numerous attempts in recent years to overcome this issue and provide architectures that aim to go beyond smoothing (e.g., MixHop, graph scattering, or various architectures with residual layers or skip connections come to mind). It seems the proposed approach here would still rely on local smoothing, and therefore it is not entirely clear if it will really contribute significantly compared to the current trends and state of the art in the field.

Therefore, I would consider this work as a borderline case, but I am leaning towards acceptance due to the contribution towards a more theoretically oriented (rather than algorithmic) unified formulation encompassing a plethora of architectures that fall under the local smoothing paradigm, while noting the paper would be more convincing if it also considered how to advance beyond this smoothing or denoising regime.

Minor remarks:
- In section 5, the authors say "GAT unintentionally adopts an adaptive C" - I would argue this is rather deliberate and not unintentional.
- The use of *author (year)* and *(author, year)* citation styles (citet and citep natbib commands) seems somewhat inconsistent, and does not follow the recommended guidelines of when to utilize each style.

---

> ### Author Response · Authors · 2020-11-18
> **Response to AnonReviewer5**
>
> We appreciate the reviewer's perception of our contributions and thank the reviewer for the insightful questions.  Our detailed responses are below.
> ***
> Q1. There is little discussion of numerous attempts in recent years to overcome this issue and provide architectures that aim to go beyond smoothing (e.g., MixHop, graph scattering, or various architectures with residual layers or skip connections come to mind). It seems the proposed approach here would still rely on local smoothing, and therefore it is not entirely clear if it will really contribute significantly compared to the current trends and state of the art in the field.
>
> A1. While our focus in this work is not the oversmoothing issue of GNN models, the unified framework we propose can actually cover some of the methods designed to overcome this issue. For example, PairNorm [1] (as mentioned in Section 4, with details in Appendix B), originally proposed as a remedy for oversmoothing, corresponds to the graph signal denoising problem with the following specific regularization term $R_A+ R_B$ with $R_A$ and $R_B$ as follows.
> $R_A = \\sum\\limits_{(i,j) \\in \\mathcal{E}} c_p \\cdot \\left\\| {\bf F}[i,:] - {\\bf F}[j,:]\\right\\|_2^2 $ and  $R_B = - \\sum\\limits_{(i,j)\\not\\in \\mathcal{E}} c_n \\cdot \\| {\\bf F}[i,:] - {\\bf F}[j,:]\\|_2^2,$
> where $c_n$ and $c_p$ are positive scalars. This regularization term not only enforces local smoothness ($R_A$), but also encourages distant nodes to have dissimilar features ($R_B$). Furthermore, the residual connection architecture mentioned by the reviewer is also covered by our framework: specifically, in Theorem 3, if we set the step size to $b=\frac{1}{c}$, then we have  ${\bf F}\leftarrow \frac{1}{2}{\bf X} +\frac{1}{2} \tilde{\bf A}{\bf X}$ (see Eq.(14) in the updated paper), which can be regarded as a sort of residual connection.  The explicit connection between MixHop and graph scattering to our proposed framework is unclear currently, so we defer these to future analysis.
> Also, note that the proposed framework can go beyond local smoothing, depending on the choice of the regularization term (ref. Problem 2).  For example, as discussed above, in PairNorm, a regularization term beyond local smoothing is introduced. Correspondingly, PairNorm goes beyond local smoothing. In general, non-local smoothing operations can be incorporated by augmenting the regularization term with associated non-local smoothness terms.
>
> Finally, we believe that the proposed framework has the potential to foster more novel GNNs, especially when we have some kind of prior knowledge about the given data, which we can incorporate into the regularization term.  For example, the Laplacian regularization term corresponds to the prior knowledge that the ground truth graph signal (and node labels) are smooth over the graph (or roughly "connected nodes tend to be similar''). However, if we believe that the node labels are piece-wise constant (i.e., there are some abrupt changes in the margin of clusters with different labels; this is actually quite common in many graph datasets), then, according to [2], graph trend filtering, which adopts $L_1$ based regularization terms, can adapt to this situation better than Laplacian regularization. Hence, an $L_1$-based regularization term such as $\sum\limits_{i\in \mathcal{V}} \sum\limits_{j\in \tilde{\mathcal{N}}(i)} \|{\bf F}[i,:]-{\bf F}[j,:] \|_1 $ and $\|{\bf L}{\bf F}\|_1$ might be more suitable for node classification task under this setting, and we can use them to design a potentially better GNN model, which is suitable for datasets with this kind of prior knowledge.
>
> The above discussion demonstrates the breadth of existing works which can fall under a denoising perspective with various regularizers; we believe the framework can be used to design even more regularizers suitable for different assumed problem settings and applications.
>
> [1] Zhao, Lingxiao, and Leman Akoglu. PairNorm: Tackling Oversmoothing in GNNs'', ICLR'2020.
> [2] Wang, Yu-Xiang, et al. "Trend Filtering on Graphs." The Journal of Machine Learning Research 17.1 (2016): 3651-3691.
> ***
> Q2. In section 5, the authors say "GAT unintentionally adopts an adaptive C" - I would argue this is rather deliberate and not unintentional.
>
> A2. By "unintentionally," we meant that in the original GAT paper, the authors did not make the connection with the graph signal denoising problem and thus did not explicitly consider adaptive $c$ as a solution to facilitate varying local smoothness. We agree that the word ``unintentionally'' could introduce confusion, and we adjusted it in the revised draft.
> ***
> Q3. The use of author (year) and (author, year) citation styles (citet and citep natbib commands) seems somewhat inconsistent, and does not follow the recommended guidelines of when to utilize each style.
>
> A3. Thanks for pointing this out; we made a thorough pass to modify the citation styles, and it should now be consistent.

---

### Decision · Program_Chairs · 2021-01-07
**Final Decision**

**Decision:**

Reject

**Comment:**

The paper argues that GNNs can be understood as a graph signal denoising. While this interpretation is not surprising and not novel, the unified view does seem insightful according to some reviewers. Yet, it is not clear how much insight can be drawn from the presented theory, as no significantly better architecture or experimental results are presented.

Additional criticism was raised wrt unclear relation between GAT and the graph signal denoising, the fact that analysis focused on one layer and does not explain the relations between layers and how nonlinear activation functions affect these theoretical findings, and that the objective of GNN cannot be viewed as a simple combination of graph denoising problems. Several reviewers complained that the paper is hard to follow.

In light of the above, despite the significant efforts of the authors to address these issues in the rebuttal, we believe the paper is below the bar and recommend Rejection.